# Triangle Distance IoU Loss, Attention-Weighted Feature Pyramid Network, and Rotated-SARShip Dataset for Arbitrary-Oriented SAR Ship Detection

**Zhijing Xu** [1] 📷, **Rui Gao** [1,*] 📷, **Kan Huang** [1] and **Qihui Xu** [2]

1   College of Information Engineering, Shanghai Maritime University, Shanghai 201306, China
2   School of Communication & Information Engineering, Shanghai University, Shanghai 200444, China
*   Correspondence: 202030310004@stu.shmtu.edu.cn; Tel.: +86-198-2173-5586

**Abstract:** In synthetic aperture radar (SAR) images, ship targets are characterized by varying scales, large aspect ratios, dense arrangements, and arbitrary orientations. Current horizontal and rotation detectors fail to accurately recognize and locate ships due to the limitations of loss function, network structure, and training data. To overcome the challenge, we propose a unified framework combining triangle distance IoU loss (TDIoU loss), an attention-weighted feature pyramid network (AW-FPN), and a Rotated-SARShip dataset (RSSD) for arbitrary-oriented SAR ship detection. First, we propose a TDIoU loss as an effective solution to the loss-metric inconsistency and boundary discontinuity in rotated bounding box regression. Unlike recently released approximate rotational IoU losses, we derive a differentiable rotational IoU algorithm to enable back-propagation of the IoU loss layer, and we design a novel penalty term based on triangle distance to generate a more precise bounding box while accelerating convergence. Secondly, considering the shortage of feature fusion networks in connection pathways and fusion methods, AW-FPN combines multiple skip-scale connections and attention-weighted feature fusion (AWF) mechanism, enabling high-quality semantic interactions and soft feature selections between features of different resolutions and scales. Finally, to address the limitations of existing SAR ship datasets, such as insufficient samples, small image sizes, and improper annotations, we construct a challenging RSSD to facilitate research on rotated ship detection in complex SAR scenes. As a plug-and-play scheme, our TDIoU loss and AW-FPN can be easily embedded into existing rotation detectors with stable performance improvements. Experiments show that our approach achieves 89.18% and 95.16% AP on two SAR image datasets, RSSD and SSDD, respectively, and 90.71% AP on the aerial image dataset, HRSC2016, significantly outperforming the state-of-the-art methods.

**Keywords:** synthetic aperture radar (SAR) image; arbitrary-oriented ship detection; differentiable rotational IoU algorithm; triangle distance IoU loss; attention-weighted feature pyramid network; multiple skip-scale connections; attention-weighted feature fusion; Rotated-SARShip dataset (RSSD)

## 1. Introduction

As an active microwave sensor, synthetic aperture radar (SAR) enables all-day, all-weather, and long-distance space-to-Earth observation without being limited by light and climate conditions [1]. With the development of spaceborne SAR high-resolution imaging technology, ship detection in SAR images has become a current research hotspot [2–8].

In recent years, with the breakthrough of convolutional neural networks (CNNs) [9] in computer vision, CNN-based methods have been introduced into SAR ship detection [10–15]. Though these works have promoted the development of this field to some extent, most of them simply apply the horizontal bounding box (HBB)-based methods used in natural scenes to SAR scenes, which still encounter severe challenges, stated as follows:

1. **Complexity of SAR scenes**—since SAR images are taken from a bird's eye perspectives, they contain diverse and intricate spatial patterns. As shown in Figure 1a, instances of small ships tend to be overwhelmed by complex inshore scenes, which inevitably interferes with the recognition of foreground objects, making it difficult for HBB-based methods to accurately distinguish ships from other background components;

2. **Diversity of ship distribution**—in SAR images, ship targets are characterized by varying scales, large aspect ratios, dense arrangements, and arbitrary orientations. In Figure 1b, the HBBs of ships with tilt angles and large aspect ratios contain considerable redundant areas, which introduce background clutter. Moreover, two HBBs of densely arranged ships have a high intersection-over-union (IoU), which is not conducive to non-maximum suppression (NMS), leading to missed detection [16].

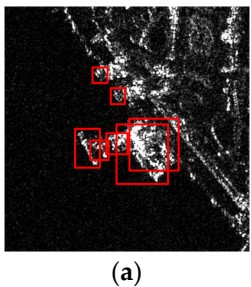 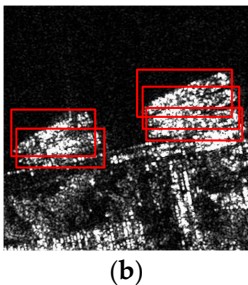 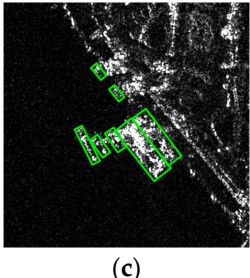 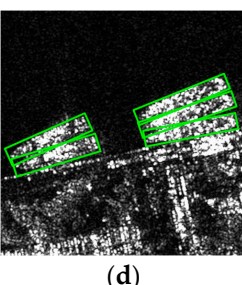

| (a) | (b) | (c) | (d) |

**Figure 1.** Densely arranged ships in complex inshore scenes. Here, (**a**,**b**) show the detecting of ship targets using the HBB-based RetinaNet [17]; (**c**,**d**) show the detecting of ship targets using the OBB-based RetinaNet with the proposed TDIoU loss and AW-FPN. The red and green boxes denote the detection results.

To eliminate the defects of HBB-based methods in detecting ships in SAR scenes, oriented bounding box (OBB)-based methods have emerged [18–22]. As shown in Figure 1c,d, OBBs can effectively avoid overlap and attenuate the influence of background clutter, enabling more precise prediction of the location and orientation of ships.

However, OBB-based methods still have the following limitations in SAR scenes:

1. **Problems of rotation detectors based on angle regression**—most rotation detectors adopt $l_n$-norms as the regression loss in the training phase and intersection-over-union (IoU) as the evaluation metric in the test phase, which will lead to loss-metric inconsistency. In addition, due to the periodicity of the angle parameter, regression-based rotation detectors usually suffer from angular boundary discontinuity [23];

2. **Constraints of multi-scale feature fusion**—due to the large variation in the shapes and scales of ship targets in SAR images, the conventional feature fusion networks [24–27], which are limited by their connection pathways and fusion methods, are not effective in detecting ships with large aspect ratios or small sizes;

3. **Deficiencies of existing SAR ship datasets**—the vast majority of SAR ship detection datasets [28–33] are still annotated by horizontal bounding boxes. Meanwhile, with potential drawbacks, such as insufficient samples, small image sizes, and relatively simple scenes, in these datasets, relevant research is hindered.

To overcome these bottlenecks, we propose a unified framework for rotated SAR ship detection. Inspired by IoU-based losses in horizontal detection, we develop a triangle distance IoU loss (TDIoU loss) and implement the forward and backward processes to ensure its trainability. Thanks to its well-designed penalty term, TDIoU loss not only solves the problems caused by angle regression but also dramatically improves convergence speed and simplifies computation. Second, to enables more effective multi-scale feature fusion for detecting ships with large aspect ratios and varying scales in complex SAR scenes, an attention-weighted feature pyramid network (AW-FPN) combining multiple skip-scale connections and the attention-weighted feature fusion (AWF) mechanism is proposed.

Finally, to promote further research in this field, a novel dataset, the rotated-SARShip dataset (RSSD), is released to provide a challenging benchmark for arbitrary-oriented ship detection in SAR images. Extensive experiments and visual analysis on three datasets prove that our approach achieves better detection accuracy than other advanced methods.

To sum up, the main contributions of this paper are summarized as follows:

1. To the best of our knowledge, TDIoU loss is the first IoU loss specifically for rotated bounding box regression. To solve the non-differentiable problem of rotational IoU, we derive an algorithm based on the Shoelace formula and implement back-propagation for it. The TDIoU loss aligns the training target with the evaluation metric and is immune to boundary discontinuity by measuring the sampling point distance and the triangle distance between OBBs without directly introducing the angle parameter. Furthermore, it is still informative for learning even when there is no overlap between two OBBs or they are in an inclusion relationship, a common occurrence in small ship detection;

2. Our AW-FPN outperforms previous methods in both connection pathways and fusion methods. Skip-scale connections inject more abundant semantic and location information into multi-scale features, facilitating the recognition and localization of ships. The AWF mechanism generates non-linear fusion weights of the same size as the input feature via a multi-scale channel attention module (MCAM) and multi-scale spatial attention module (MSAM), enabling soft feature selections in an element-wise manner, which is critical for detecting ships with large aspect ratios or small sizes;

3. We construct a large-scale RSSD for detecting ships with arbitrary orientations and large aspect ratios in SAR images. To ensure data diversity, we collect original images from three SAR satellites and select different imaging areas. With the help of the automatic identification system (AIS) and Google Earth, 8013 SAR images, including 21,479 ships, are precisely annotated by rotated ground truths. Moreover, we conduct comprehensive statistical analysis and provide results of 15 baseline methods on our dataset. Notably, RSSD is the largest current dataset for rotated SAR ship detection;

4. We embed TDIoU loss and AW-FPN as plug-ins into baseline models and conduct comparative experiments with a dozen popular rotation detectors on two SAR image datasets, the RSSD and the SSDD, and one aerial image dataset, HRSC2016. The results prove that our approach not only achieves state-of-the-art performance in SAR scenes, but also that it shows excellent generalization ability in optical remote sensing scenes.

The rest of the paper is organized as follows: Section 2 reviews related works. Section 3 describes the problems in angle regression and conventional IoU-based losses. Section 4 introduces the proposed TDIoU loss and the AW-FPN for rotated SAR ship detection. Section 5 presents details of the proposed RSSD. Extensive experiments and comprehensive discussions are provided in Section 6. Section 7 summarizes the whole work.

## 2. Related Work

In this section, we first review CNN-based SAR ship detection methods, then discuss the related works dealing with the problems caused by angle regression and multi-scale feature fusion, and finally analyze several existing publicly available SAR ship datasets.

### 2.1. SAR Ship Detection Methods Based on Convolutional Neural Networks

In the field of object detection, convolutional neural networks have become the mainstream algorithm. In recently years, CNN-based methods have made significant progress in SAR ship detection. As a pioneering work, Li et al. [10] discussed the defects of Faster R-CNN [34] in SAR ship detection and proposed an improved framework based on feature fusion and hard negative mining. Zhang et al. [11] proposed a novel concept of balance learning (BL) for high-quality SAR ship detection. Zhang et al. [12] proposed a grid convolutional network with depthwise separable convolution that accelerates ship detection by griding the input image. To enhance the detailed features of ships, Liang et al. [13] proposed a visual attention mechanism. Furthermore, the means dichotomy method and speed block

kernel density estimation method were used for adaptive hierarchical ship detection. Gao et al. [14] achieved better ship detection accuracy by using the anchor-free CenterNet [35] based on an attention mechanism and feature reuse strategy. Zhang et al. [15] designed a quad feature pyramid network consisting of four unique FPNs and verified its effectiveness on five SAR datasets.

However, the above methods fail to take into account the large aspect ratio and multi-angle characteristics of ships, leading to missed and false detection. Therefore, in recent years, there has been some research on rotated ship detection. For instance, Wang et al. [18] added the angle regression and semantic aggregation method to SSD. The attention module was used to adaptively select meaningful features of ships. Chen et al. [19] presented a feature-guided alignment module and a lightweight non-local attention module to balance the detection accuracy and inference speed of single-stage rotation detectors. Pan et al. [16] constructed a multi-stage rotational region-based network that generates rotated anchors through a rotation-angle-dependent strategy. To reduce the false alarm rate, Yang et al. [20] devised a novel loss to balance the loss contribution of various negative samples. To enhance the detection of small ships, An et al. [21] proposed an anchor-free rotation detector with a flexible frame. Sun et al. [22] applied the bi-directional feature fusion module and angle classification technique to a YOLO-based rotated ship detector.

### 2.2. Loss-Metric Inconsistency and Angular Boundary Discontinuity

To eliminate the gap between the bounding box regression loss and the evaluation metric, IoU-based losses have been introduced in horizontal detectors [36–40]. Unfortunately, they cannot be simply applied to rotation detection, as the general rotational IoU algorithm is non-differentiable for back-propagation. In addition, unlike other bounding box parameters, the angle parameter is periodic in nature, which will lead to a surge in loss value at the boundary of the angle definition range when using $l_n$-norm losses.

Some studies have attempted to address part of the above issues from two perspectives. One idea is to design differentiable approximate IoU losses for angle regression. To control the loss value by the amplitude of IoU, Yang et al. [41] added an extra IoU factor into the smooth L1 loss. Furthermore, PIoU [42] estimated the intersection area of two rotated bounding boxes by roughly counting the number of pixels. Aiming to address the uncertainty of convex shapes, Zheng et al. [43] presented an affine transformation to estimate the intersection area. The GWD [23] converted the oriented bounding box to two-dimensional Gaussian distribution, using the Gaussian–Wasserstein distance to approximate the rotational IoU loss. Although these improved regression losses alleviate the problems to some extent, their gradient directions are still not dominated by IoU, and they cannot accurately guide training.

Another idea is to treat the angle prediction as a discrete classification task so as to properly constrain the prediction results. Yang et al. [44] developed a circular smooth label (CSL) technique that directly uses the angle parameter as the category label to tackle the periodicity of the angle and improve the tolerance of adjacent angles. The DCL [45] analyzed the problems of over-thick prediction heads in sparse coded labels and converted the angle categories into dense codes, such as the binary codes and gray codes, to further improve the detection efficiency. Although angle classification techniques avoid angular boundary discontinuity, they are still limited by angular discretization granularity, which inevitably leads to theoretical errors in high-precision angle prediction.

As of now, no full-fledged method exists to address all the above issues. In a sense, the proposed differentiable rotational IoU algorithm opens up the possibility of using the IoU-based loss for rotated bounding box regression, and the newly designed TDIoU loss fundamentally eliminates all these problems in an ingenious manner.

### 2.3. Multi-Scale Feature Fusion

In CNNs, high-level features contain richer semantic information and broader receptive fields, making them beneficial for detecting large ship targets. Low-level features are of

high resolution and contain abundant shallow information, which is conducive to locating small ship targets. One of the difficulties in SAR ship detection is how to effectively fuse multi-scale features. Figure 2 displays several mainstream feature fusion networks [24–27]. Analysis shows that they still suffer from the following limitations in SAR scenes:

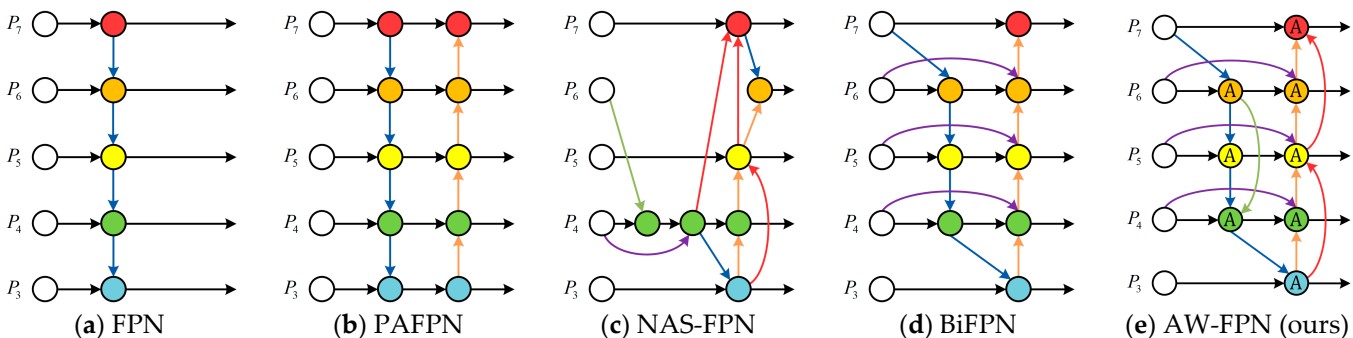

**Figure 2.** Feature fusion networks. Here, $P_i$ indicates the feature pyramid level $_i$. (**a**) The FPN proposes a top-down pathway to fuse multi-scale features from $P_3$ to $P_7$ ; (**b**) PANet builds up an extra bottom-up pathway; (**c**) NAS-FPN designs the network topology by the neural architecturesearch; (**d**) BiFPN adds transverse skip-scale connections and learnable scalar fusion weights; (**e**) our AW-FPN with multiple skip-scale connections and attention-weighted feature fusion (AWF) mechanism.

1.  **Restricted connection pathway**—the conventional feature pyramid network (FPN) [24] is inherently limited by a single top-down information flow. Therefore, in PANet [25], an extra bottom-up path aggregation network is added. The above two methods only consider adjacent-level feature fusion. To solve this problem, BiFPN [27] added transverse skip-scale connections from input nodes to output nodes. However, such single same-level feature reuse ignores semantic interactions between cross-level features. Due to the relatively long pathways between high-level features and low-level features, semantics are likely be weakened during layer-to-layer transmission, which is not conducive to the detection of ships with extreme shapes and scales;
2.  **Inappropriate fusion method**—most works on feature fusion focus only on designing complicated connection pathways. The fusion method, usually realized by simple addition, is rarely mentioned. Due to the different resolutions of different feature levels, their contributions to the output should also be unequal. The BiFPN added learnable scalar weights to the input features of each node. However, such a rough weighting method, which makes no distinction between all feature points, is still a linear combination of feature maps. Since ship targets in the same image usually have significant differences in scale, simple linear aggregation might not be the best choice.

In recent years, several investigations on visual attention have begun to focus on the fusion method. In SKNet [46] and ResNest [47], the global channel attention mechanism [48] is used to conduct dynamic weighted averaging of features from multiple kernels or groups. Although these attention-based approaches achieve non-linear feature fusion, they only show solicitude for the feature selections in the same layer, leaving no solution for fusing cross-level features of inconsistent semantics and scales. Furthermore, global channel attention only generates a scalar fusion weight for each channel of the feature map, which is obviously not appropriate for scenes with large variations in target scale. Generally speaking, multi-scale networks need to learn diverse feature representations, and a single global channel interaction will weaken the context information of small targets. Recently, aiming to provide a paradigm for cross-level feature fusion, Dai et al. [49] proposed an attentional feature fusion (AFF) mechanism. Regrettably, as with previous approaches, AFF only tends to focus on the salience representations of features in the channel dimension, which might result in the loss of multi-scale spatial contexts.

Our AW-FPN has improved on both of the above. To enrich the semantic and location information in feature maps, both transverse and longitudinal skip-scale connections are

used. To generate high-quality fusion weights, a novel AWF mechanism is proposed. The MCAM and MSAM in AWF aggregate both multi-scale channel and spatial contexts, so as to emphasize the region around real ship targets and suppress background clutter.

### 2.4. SAR Image Datasets for Ship Detection

Due to the limitations of SAR imaging conditions, the datasets of SAR scenes are not as diverse as those of natural scenes. Recent research has been committed to constructing larger and more comprehensive SAR ship detection datasets. Table 1 shows the statistics of six existing datasets [28–33]. However, they still suffer from the following defects:

1.  **Insufficient training samples**—the existing SAR ship datasets, such as SSDD [28], DSSDD [30], and AIR-SARShip [31], have a relatively small number of image samples and, therefore, require a large amount of data augmentation before training, which is not conducive to training a high-precision ship detection network;

2.  **Small image sizes and relatively simple scenes**—in the SAR-Ship-Dataset [29], ship slices are only 256 × 256 pixels in size. As a matter of fact, small ship slices are more suitable for ship classification since they contain simpler scene information and less inshore scattering. As a result, detectors trained on these ship slices may have difficulty in locating ships near highly reflective objects in large-scale scenes [32];

3.  **Inappropriate annotations**—most existing datasets in this field, which fail to consider the large aspect ratio and multi-angle characteristics of ships, are still annotated by HBBs without shape and orientation information. In contrast, OBBs can better fit the approximate shape of ships and mitigate the effect of background clutter. Notably, HRSID [32] and SSDD adopt the polygon annotation for ship instance segmentation. Semantic segmentation divides each pixel of an image into a semantically interpretable class and highlights instances of the same class with the same color. On this basis, instance segmentation employs the results of object detection to perform an instance-level segmentation on different targets of the same class. Although segmented polygons generated by pixel-wise masks enable more accurate contour detection, they are costly in both annotation and detection. For ships in SAR images, we prefer to learn about their general shapes, such as aspect ratio and orientation. On balance, the OBB annotation is a relatively suitable choice. So far, only SSDD provides OBB annotations. However, it contains only 1160 images with 2587 ships, which is far from meeting the demands of ship detection in complex SAR scenes. Hence, it is necessary to construct a large-scale dataset specifically for arbitrary-oriented SAR ship detection.

**Table 1.** Statistics of the six SAR ship detection datasets released in references [28–33] and our proposed RSSD.

| Datasets | Satellite | Polarization | Resolution (m) | Image Size (Pixel) | Image Number | Ship Number | Annotations |
|---|---|---|---|---|---|---|---|
| SSDD [28] | RadarSat-2, TerraSAR-X, Sentinel-1 | HH, HV, VV, VH | 1~15 | (214~653) × (190~526) | 1160 | 2587 | HBB, OBB, Polygon |
| SAR-Ship-Dataset [29] | Gaofen-3, Sentinel-1 | Single, Dual, Ful | 3, 5, 8, 10, 25, etc. | 256 × 256 | 43,819 | 59,535 | HBB |
| DSSDD [30] | RadarSat-2, TerraSAR-X, Sentinel-1 | – | 1~5 | 416 × 416 | 1174 | – | HBB |
| AIR-SARShip [31] | Gaofen-3 | Single, VV | 1, 3 | 1000 × 1000, 3000 × 3000 | 331 | – | HBB |
| HRSID [32] | Sentinel-1, TerraSAR-X | HH, HV, VV | 0.5, 1, 3 | 800 × 800 | 5604 | 16,951 | HBB, Polygon |
| LS-SSDD-v1.0 [33] | Sentinel-1 | VV, VH | 5 × 20 | about 24,000 × 16,000 | 15 | 6015 | HBB |
| **RSSD (ours)** | **Sentinel-1, TerraSAR-X, Gaofen-3** | **Single, HH, HV, VV** | **0.5, 1, 3, 5 × 20** | **800 × 800** | **8013** | **21,479** | **OBB** |

Our proposed RSSD acquires data from three SAR satellites with different resolutions, polarizations, and imaging modes. The imaging areas are selected in ports and canals with busy trade. All images have been meticulously pre-processed and split into 8013 ship slices of 800 × 800 pixels. With the help of professional tools, 21,479 ships are precisely annotated by OBBs. All these treatments contribute to the complexity and diversity of our dataset.

## 3. Analysis of Angle Regression Problems and Conventional IoU-Based Losses

In this section, we first discuss two major problems in the existing rotation detectors mainly caused by angle regression. Then, we review the conventional IoU-based losses and analyze the limitations they may encounter in rotated bounding box regression. Finally, we summarize several requirements that should be met for the rotational IoU loss.

### 3.1. Problems of Rotation Detectors Based on Angle Regression

Figure 3 demonstrates two generic parametric definitions of oriented bounding boxes (i.e., OpenCV definition and long-edge definition). According to the above two definitions, any two-dimensional bounding box can be represented as a group of five parameters ($cx$, $cy$, $w$, $h$, and $\theta$), where ($cx$, $cy$) represents the centroid coordinate of the oriented bounding box, $w$ and $h$ indicate the width and height, respectively, and $\theta$ denotes the rotation angle. To predict the angle $\theta$ of the bounding box, most rotation detectors directly introduce an additional output channel into the regression subnet and use $l_n$-norms as the regression loss during the training phase. However, in the testing stage, the performance is evaluated by IoU. Obviously, such a mismatch may present some problems, which we will now summarize.

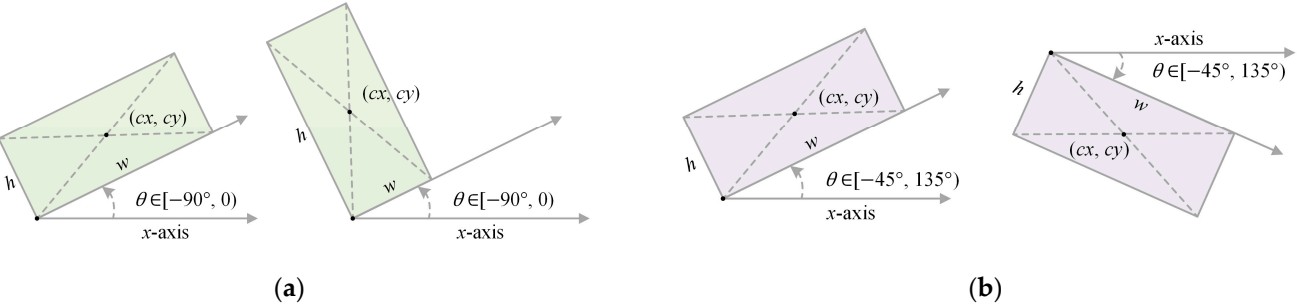

(**a**)                                                                                    (**b**)

**Figure 3.** Two generic parametric definitions of oriented bounding boxes. (**a**) OpenCV definition, where $\theta$ indicates the acute or right angle between the width $w$ and the $x$-axis; (**b**) Long-edge definition, where $w$ and $h$ signify the long side and short side of a bounding box, respectively. Here, $\theta$ denotes the angle from the $x$-axis to the direction of the width $w$.

### 3.1.1. Loss-Metric Inconsistency

In Figure 4a, we compare the relationships between different regression losses and angle differences. Despite the fact that they are all monotonic, only the IoU loss (the light blue curve) and our TDIoU loss (the navy blue curve) are concave, indicating that the gradient directions of $l_n$-norms are inconsistent with that of IoU. Figure 4b displays the relationship between the rotational IoU and angle differences under different aspect ratios. For a target with a large aspect ratio, a slight angle difference will also lead to a rapid drop in the IoU value. Figure 4c displays the relationships between different regression losses and aspect ratios. All $l_n$-norm losses remain constant regardless of aspect ratio variations, while the IoU-based losses vary dramatically. The loss-metric inconsistency leads to the conclusion that even a small training loss cannot guarantee high detection performance.

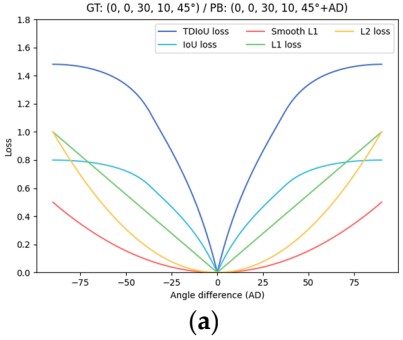 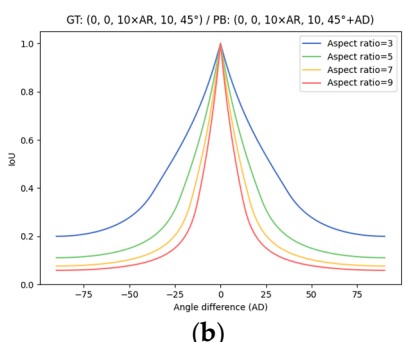 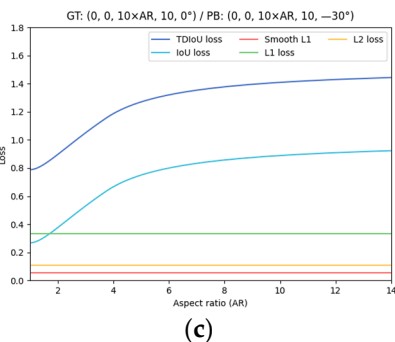

(**a**)         (**b**)         (**c**)

**Figure 4.** Loss-metric inconsistency. All ground truths (GT) and predicted boxes (PB) are represented as ($cx$, $cy$, $w$, $h$, and $\theta$) under the long-edge definition. (**a**) Regression loss variations versus angle differences (AD); (**b**) rotational IoU variations versus angle differences under different aspect ratios (AR); (**c**) regression loss variations versus aspect ratios.

### 3.1.2. Angular Boundary Discontinuity

The angular boundary discontinuity refers to the surge in loss at the boundary of the angle definition range due to the periodicity of the angle (PoA) and the exchangeability of edges (EoE) [23]. Figure 5a shows the boundary problem under the OpenCV definition. Suppose there is a blue anchor/proposal and a green ground truth. The angle of the anchor/proposal is exactly around the maximum or minimum of the defined range. The ideal regression form is to rotate the anchor/proposal counterclockwise by a small angle to the position of the red box. However, due to the angle periodicity, the angle of the predicted box exceeds the defined range $[-90°, 0)$, and the width and height are interchanged relative to the ground truth, leading to a large smooth L1 loss. At this point, the anchor/proposal has to be regressed in a more complex way. For example, it should be rotated clockwise by a larger angle, and its width and height should be scaled at the same time. A similar phenomenon also occurs under the long-edge definition, as shown in Figure 5b.

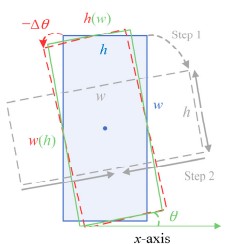 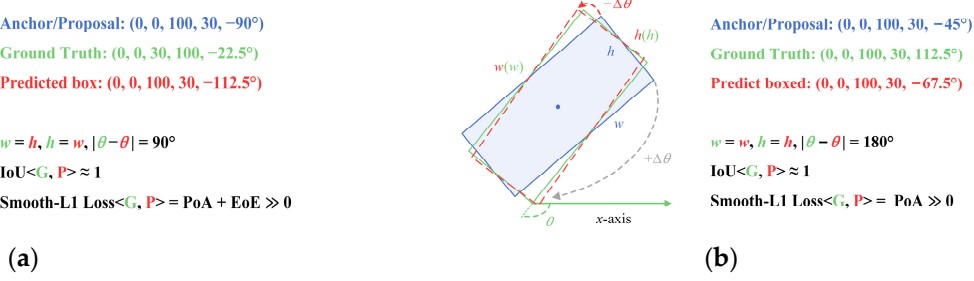

(**a**)                                               (**b**)

**Figure 5.** Angular boundary discontinuity under (**a**) the OpenCV definition and (**b**) the long-edge definition.

In essence, angular boundary discontinuity is a kind of manifestation of loss-metric inconsistency. In the boundary case, even if the IoU between the predicted box and the ground truth is very high, a considerable loss will be incurred. Based on the above analysis, we can conclude that the $l_n$-norms are inapplicable to rotated bounding box regression.

### 3.2. Limitations of Conventional IoU-Based Losses

It has been demonstrated in horizontal detection methods that the IoU-based losses [36–40] can ensure that the training target remains consistent with the evaluation metric. In theory, they should also work in the rotation case, as the only difference is that the IoU computation for oriented bounding boxes is more complex than that for horizontal ones.

Compared to $l_n$-norms, the IoU loss has several merits. Firstly, the IoU computation involves all of the geometric properties of bounding boxes, including location, orientation, shape, etc. Secondly, instead of treating the parameters as independent variables as in

the case of $l_n$-norms, IoU implicitly encodes the relationship between each parameter by area calculation. Finally, IoU is scale-invariant, making it ideal for solving scale and range disparities between individual parameters. The original IoU loss is defined as follows [37]:

$$L_{IoU} = 1 - \text{IoU} \tag{1}$$

Here, $L_{IoU}$ is valid only when two bounding boxes have overlap and would not offer any moving gradient for non-overlapping cases. Moreover, it cannot reflect the manner in which the boxes intersect. In Figure 6, the relative positions between the predicted box and the ground truth are obviously different, while the evaluation results of $L_{IoU}$ remain constant.

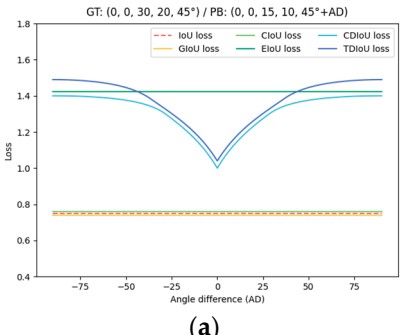

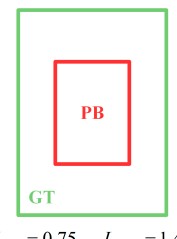
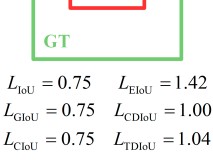

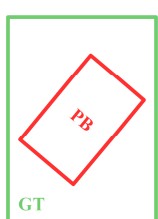
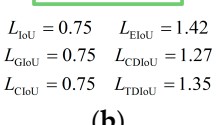

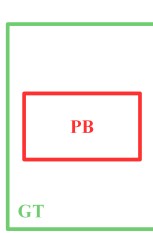

(**a**)                                        (**b**)

**Figure 6.** Comparison between different IoU-based losses. (**a**) Different IoU-based loss curves versus angle differences; (**b**) some examples from (**a**). When $B^{pb}$ and $B^{gt}$ with coincident centroids are in a containment relationship and their widths and heights are constant, GIoU loss, CIoU loss, and EIoU loss all degenerate into the original IoU loss. In contrast, our TDIoU loss (the navy blue curve) is still able to stably reflect the angle difference and is informative for learning.

The GIoU loss [37] alleviates the issue of gradient disappearance in the non-overlapping case by adding an additional penalty term, which is expressed as follows:

$$L_{GIoU} = 1 - \text{IoU} + \frac{\left| C - B^{pb} \cup B^{gt} \right|}{|C|} \tag{2}$$

where $B^{pb}$ and $B^{gt}$ are the predicted box and the ground truth, and $C$ denotes the smallest enclosing box covering $B^{pb}$ and $B^{gt}$. Research shows that GIoU first tries to increase the size of $B^{pb}$ to overlap $B^{gt}$ and then uses the IoU term to maximize the intersection area of the bounding boxes [40]. Moreover, GIoU loss requires more iterations to converge.

When designing the penalty term, CIoU loss [38] takes into account the centroid distance and the aspect ratio of the bounding boxes, which is defined as follows:

$$L_{CIoU} = 1 - \text{IoU} + \frac{\rho^2\left( b^{pb},\, b^{gt} \right)}{c^2} + \alpha v \tag{3}$$

$$v = \frac{4}{\pi^2}\left( \arctan\frac{w^{gt}}{h^{gt}} - \arctan\frac{w^{pb}}{h^{pb}} \right),\ \alpha = \frac{v}{(1 - IoU) + v} \tag{4}$$

where $b^{pb}$ and $b^{gt}$ represent the centroids of $B^{pb}$ and $B^{gt}$, respectively; $\rho(\cdot)$ indicates the Euclidean distance; $c$ denotes the diagonal length of the smallest enclosing box; $w^{pb}$ and $h^{pb}$ signify the width and height of $B^{pb}$, respectively; $w^{gt}$ and $h^{gt}$ signify the width and height of $B^{gt}$, respectively. In CIoU loss, $v$ only reflects the difference in the aspect ratio, rather than the actual difference between $w^{pb}$ and $w^{gt}$ (or $h^{pb}$ and $h^{gt}$).

To solve this problem, EIoU loss [39] proposes a more efficient form of penalty term:

$$L_{EIoU} = 1 - \text{IoU} + \frac{\rho^2\left(b^{pb}, b^{gt}\right)}{c^2} + \frac{\rho^2\left(w^{pb}, w^{gt}\right)}{c_w^2} + \frac{\rho^2\left(h^{pb}, h^{gt}\right)}{c_h^2} \quad (5)$$

where $c_w$ and $c_h$ indicate the width and height of the smallest enclosing box, respectively. The EIoU loss directly minimizes the difference in the width and height between $B^{pb}$ and $B^{gt}$, leading to faster convergence and more accurate bounding box regression.

Recently, a new form of penalty term was released in CDIoU loss [40], which narrows the difference between $B^{pb}$ and $B^{gt}$ by minimizing the distance between their vertices, as follows:

$$L_{CDIoU} = 1 - \text{IoU} + \frac{B^{pb} - B^{gt}{}_2}{c^2} \quad (6)$$

where $B^{pb} - B^{gt}{}_2$ is the distance between the corresponding vertices of $B^{pb}$ and $B^{gt}$.

However, the above IoU-based losses are all designed for horizontal detection. Due to the introduction of the angle parameter, applying them to oriented bounding box regression will bring some problems. As shown in Figure 6a,b, when $B^{pb}$ and $B^{gt}$ with coincident centroids are in a containment relationship and their widths and heights are constant, the values of GIoU loss, CIoU loss, and EIoU loss remain the same regardless of changes in the angle of $B^{pb}$. At this point, they completely degenerate into the original IoU loss, making the regression more difficult and the convergence slower. In other words, general parameter-based penalty terms cannot effectively measure the angle difference between $B^{pb}$ and $B^{gt}$. A natural idea is to introduce the angle parameter into the penalty term. Nevertheless, such a treatment will reintroduce the angular boundary discontinuity, which goes against our original intention. In addition, we also find that the penalty term of CDIoU loss based on the vertex distance is sensitive to the angle parameter. Unfortunately, the denominator of its penalty term involves computing the smallest enclosing box covering $B^{pb}$ and $B^{gt}$, an extremely tricky task for two rotated boxes. Since the shape of the convex hull formed by the vertices of $B^{pb}$ and $B^{gt}$ is not fixed, the oriented minimum bounding box algorithm [50] requires exhaustive enumeration to obtain the final result, which will consume a lot of computing time and delay the whole training process.

To sum up, a qualified rotational IoU loss should at least meet the following four requirements:

1.  **Requirement 1**—it should be differentiable for back-propagation;
2.  **Requirement 2**—it should be continuous at the boundary of the angle definition range;
3.  **Requirement 3**—it should stably reflect the angle difference between bounding boxes;
4.  **Requirement 4**—the computation of the penalty term should be as simple as possible.

## 4. The Proposed Method

This section elaborates on our proposed unified framework for detecting arbitrary-oriented ships in SAR images, including the differentiable rotational IoU algorithm based on the Shoelace formula, the triangle distance IoU loss (TDIoU loss), and the attention-weighted feature pyramid network (AW-FPN) combining multiple skip-scale connections and the attention-weighted feature fusion (AWF) mechanism.

### 4.1. Differentiable Rotational IoU Algorithm Based on the Shoelace Formula

Figure 7 visualizes the computation of the intersection-over-union (IoU) for horizontal and oriented bounding boxes. For two-dimensional object detection, the IoU between the ground truth $B^{gt}$ and the predicted box $B^{pb}$ is defined as follows [51]:

$$\text{IoU}\left(B^{gt}, B^{pb}\right) = \frac{\left|B^{gt} \cap B^{pb}\right|}{\left|B^{gt} \cup B^{pb}\right|} = \frac{Area_{intersect}}{Area_{union}} = \frac{Area_{intersect}}{Area_{gt} + Area_{pb} - Area_{intersect}} \quad (7)$$

where $\left| B^{gt} \cap B^{pb} \right|$ and $Area_{intersect}$ signify the area of the intersection area, and $\left| B^{gt} \cup B^{pb} \right|$ and $Area_{union}$ imply the area of the union area. $Area_{gt}$ and $Area_{pb}$ denote the area of $B^{gt}$ and $B^{pb}$, respectively. It can be found that how to calculate $Area_{intersect}$ is the core issue. However, as shown in Figure 7b, the IoU computation for OBBs is more complex than that for HBBs, since the shape of the intersection area in the rotation case could be any polygon with fewer than eight edges. In addition, the general rotational IoU algorithm [52] is non-differentiable, as it uses triangulation to calculate $Area_{intersect}$. To address the above issue, we derive a differentiable rotational IoU algorithm based on the Shoelace formula [53], whose pseudo code is provided in Algorithm 1 (Pseudo code of the proposed rotational IoU algorithm based on the Shoelace formula). To further apply it to the IoU loss layer, we implement its forward and backward computation, as illustrated in Figure 8.

---

**Algorithm 1:** IoU computation for oriented bounding boxes

---

**Input:** Vertex coordinates of $B^{gt}$ and $B^{pb}$
**output:** IoU value

1:     Compute the area of $B^{gt}$ and $B^{pb}$: $Area_{gt} \leftarrow RectArea\left(B^{gt}\right)$; $Area_{pb} \leftarrow RectArea\left(B^{pb}\right)$;
2:     Get the edges of $B^{gt}$ and $B^{pb}$: $Edge_{gt} \leftarrow GetEdge\left(B^{gt}\right)$; $Edge_{pb} \leftarrow GetEdge\left(B^{pb}\right)$;
3:     Initialize $A \leftarrow 0$ and the vertices of the intersection area $V \leftarrow EmptySet$;
4:     **for** $i \leftarrow 1$ **to** 4 **do**
5:         Get the vertices of $B^{gt}$ inside $B^{pb}$: $V \leftarrow V.add\left(DotProduct\left(B^{gt}(i), B^{pb}\right)\right)$;
6:         Get the vertices of $B^{pb}$ inside $B^{gt}$: $V \leftarrow V.add\left(DotProduct\left(B^{gt}, B^{pb}(i)\right)\right)$;
7:         **for** $j \leftarrow 1$ **to** 4 **do**
8:         Get the intersection of edges: $V \leftarrow V.add\left(Bezier\left(Edge_{gt}(i), Edge_{pb}(j)\right)\right)$;
9:         **end for**
10:    **end for**
11:    Sort the vertices of the intersection area: $Indices \leftarrow SortVertex(V)$;
12:    Gather the sorted vertex coordinates according to indices: $V' \leftarrow Gather(V, Indices)$;
13:    **for** $n \leftarrow 1$ **to** $len(V)$ **do**
14:    Shoelace Formula: $A \leftarrow A + V'(n, 1) \times V'(n+1, 2) - V'(n, 2) \times V'(n+1, 1)$
15:    **end for**
16:    Compute the area of the intersection area: $Area_{intersect} \leftarrow A / 2$;
17:    **return** IoU $\leftarrow Area_{intersect} / \left(Area_{gt} + Area_{pb} - Area_{intersect}\right)$

---

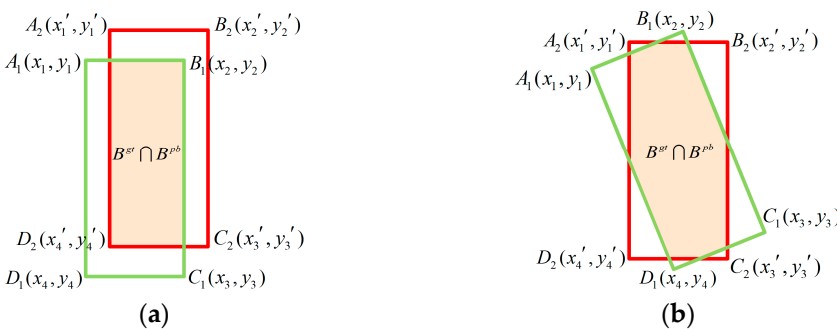

**Figure 7.** IoU computation for (**a**) horizontal and (**b**) oriented bounding boxes. Red and green boxes represent the predicted box and the ground truth, and the intersection area is highlighted in orange.

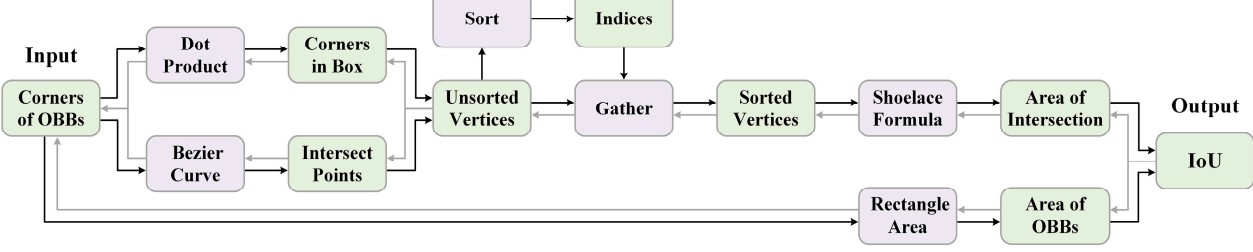

**Figure 8.** The forward and backward computation of the proposed rotational IoU algorithm. Green and purple boxes signify tensors and operators, respectively. Black and grey arrows indicate forward and backward processes, respectively.

### 4.1.1. Forward Process

On the basis of Algorithm 1 and Figure 8, the forward process is as follows:

**Step 1**—convert the ground truth $B^{gt}$ and the predicted box $B^{pb}$ into vertex coordinate representations and calculate their areas (i.e., $Area_{gt}$ and $Area_{pb}$, respectively);

**Step 2**—find the vertices of the intersection area of $B^{gt}$ and $B^{pb}$. These are located on the basis of two cases, as follows: (1) from the vertex of $B^{gt}$ and $B^{pb}$, which falls just inside the other box, and (2) from the intersection point between the edges of two rotated boxes. In the former case, we use the dot product to calculate the projection of each vertex of $B^{gt}$ and $B^{pb}$ onto two adjacent edges of the other box, respectively, and then determine whether the vertex falls inside the other box, by judging whether the projection exceeds the extent of the corresponding edge. In the latter case, since each edge of rotated boxes is a line segment defined by two vertices, the problem is transformed into locating the intersection point between two line segments in two-dimensional space [54].

Suppose $L_1$ is an edge of $B^{gt}$, defined by two vertices $(x_1, y_1)$ and $(x_2, y_2)$, and $L_2$ is an edge of $B^{pb}$, defined by two vertices $(x_3, y_3)$ and $(x_4, y_4)$. The line segments $L_1$ and $L_2$ can be defined in terms of first-degree Bezier parameters, as follows [55]:

$$L_1 = \begin{bmatrix} x_1 \\ y_1 \end{bmatrix} + t \begin{bmatrix} x_2 - x_1 \\ y_2 - y_1 \end{bmatrix}, \ L_2 = \begin{bmatrix} x_3 \\ y_4 \end{bmatrix} + u \begin{bmatrix} x_4 - x_3 \\ y_4 - y_3 \end{bmatrix} \tag{8}$$

where both $t$ and u are real numbers, and can be expressed as follows:

$$t = \frac{\det \begin{bmatrix} x_1 - x_3 & x_3 - x_4 \\ y_1 - y_3 & y_3 - y_4 \end{bmatrix}}{\det \begin{bmatrix} x_1 - x_2 & x_3 - x_4 \\ y_1 - y_2 & y_3 - y_4 \end{bmatrix}}, \ u = \frac{\det \begin{bmatrix} x_1 - x_3 & x_1 - x_2 \\ y_1 - y_3 & y_1 - y_2 \end{bmatrix}}{\det \begin{bmatrix} x_1 - x_2 & x_3 - x_4 \\ y_1 - y_2 & y_3 - y_4 \end{bmatrix}} \tag{9}$$

where $\det[\cdot]$ represents the determinant computation. If, and only if, $0 \leq t \leq 1$ and $0 \leq u \leq 1$, an intersection point $(P_x, P_y)$ exists as follows:

$$(P_x, P_y) = (x_1 + t(x_2 - x_1), \ y_1 + t(y_2 - y_1)) = (x_3 + u(x_4 - x_3), \ y_3 + u(y_4 - y_3)) \tag{10}$$

In particular, when $L_1$ and $L_2$ are collinear (parallel or coincident), they do not intersect. By traversing each edge of $B^{gt}$ and $B^{pb}$, we obtain all the intersection points.

By computing the above two cases, we finally determine the vertices of the intersection area. If the vertex does not exist, the IoU value is zero;

**Step 3**—sort the vertices of the intersection area. In general, the vertices of the intersection area form a convex hull. To compute its area, we need to sort its vertices. First, calculate the mean value of the abscissa and the ordinate of these vertices, and note it as the centroid of the polygon. Second, compute the vectors from the centroid to each vertex and normalize them to simplify the sort operation. Finally, scan all the vertices in counterclockwise order from the positive direction of the $x$-axis to obtain the sorted vertex indices.

**Step 4**—perform the gather operation to successively fetch the actual coordinate values of the sorted vertices from the unsorted vertex tensor according to the indices;

**Step 5**—compute the area of the intersection polygon using the Shoelace formula, as follows [56]:

$$Area_{intersect} = \frac{1}{2} \left| \sum_{i=1}^{n} x_i(y_{i+1} - y_{i-1}) \right| = \frac{1}{2} \left| \sum_{i=1}^{n} y_i(x_{i+1} - x_{i-1}) \right| = \frac{1}{2} \left| \sum_{i=1}^{n} \det \begin{bmatrix} x_i & x_{i+1} \\ y_i & y_{i+1} \end{bmatrix} \right| \tag{11}$$

where $n$ represents the number of edges of the intersection polygon; $(x_i, y_i)$ indicate the sorted vertices of the polygon, where $i = 1, 2, \cdots, n$. Note that $x_{n+1} = x_1$ and $y_{n+1} = y_1$;

**Step 6**—compute the rotational IoU value of $B^{gt}$ and $B^{pb}$ according to Equation (7).

4.1.2. Backward Process

During the forward process, the sort operation returns the indices of sorted vertices in counterclockwise order. Since the return value is discrete (an integer number) rather than continuous (a float number), it is non-differentiable and, therefore, cannot participate in the backward process. However, the computation part of the rotational IoU is still differentiable. This is because we only use the gather operation to obtain the coordinate values of the sorted vertices on the basis of the indices returned by the sort operation, and then adopt the Shoelace formula to compute the area of the intersection area. Throughout the process, the sort operation is not really involved in the area calculation. In most existing deep learning frameworks, the gather function is defined to gather values from the input tensor along a specified dimension and according to a specified index. As its return value is continuous by definition, it is differentiable. Furthermore, the computing process of IoU, including the dot product, the line–line intersection algorithm, and the Shoelace formula, only comprises some essential additive and multiplicative operations, ensuring that the process is robust to the rotational case and feasible for back-propagation.

*4.2. Triangle Distance IoU Loss*

The proposed rotational IoU algorithm enables back-propagation of the IoU loss layer and, thus, meets **Requirement 1**. In this part, we aim to design a rotational IoU-based loss, which fulfills **Requirements 2, 3, and 4** by constructing a proper penalty term.

Similarly to [37], we define the IoU-based loss as follows:

$$L = 1 - \text{IoU} + \mathcal{R}\left(B^{pb}, B^{gt}\right) \tag{12}$$

where $\mathcal{R}\left(B^{pb}, B^{gt}\right)$ is the penalty term for the predicted box $B^{pb}$ and the ground truth $B^{gt}$.

Inspired by CDIoU, we apply the distance between corresponding sampling points (i.e., centroids and vertices) of $B^{gt}$ and $B^{pb}$ to the penalty term to measure the overall similarity between them, while avoiding the angular boundary discontinuity caused by the direct introduction of the angle parameter. To reduce the computing complexity, a novel reference term, namely triangle distance, is devised as the denominator of the penalty term to replace the diagonal length of the smallest enclosing box. Following this idea, we design a triangle distance IoU loss (TDIoU loss), which is defined as follows:

$$L_{TDIoU} = 1 - \text{IoU} + \mathcal{R}_{TDIoU} \tag{13}$$

According to Figure 9a, the penalty term of TDIoU loss is defined as follows:

$$\mathcal{R}_{TDIoU} = \frac{|AE| + |BF| + |CG| + |DH| + |PQ|}{\left|\Delta_{AEQ}^{AQ,EQ}\right| + \left|\Delta_{BFQ}^{BQ,FQ}\right| + \left|\Delta_{CGQ}^{CQ,GQ}\right| + \left|\Delta_{DHQ}^{DQ,HQ}\right| + \left|\Delta_{APQ}^{AP,AQ}\right|} \tag{14}$$

where *ABCD* and *EFGH* indicate the corresponding vertices of the predicted box $B^{pb}$ and the ground truth $B^{gt}$. Here, *P* and Q represent the centroids of $B^{pb}$ and $B^{gt}$, respectively. Furthermore, $|\cdot|$ refers to the Euclidean distance between two sampling points, while $\left|\Delta_{AEQ}^{AQ,EQ}\right|$ indicates the sum of the two edges *AQ* and *EQ* of $\Delta AEQ$ (the same applies for other similar terms).

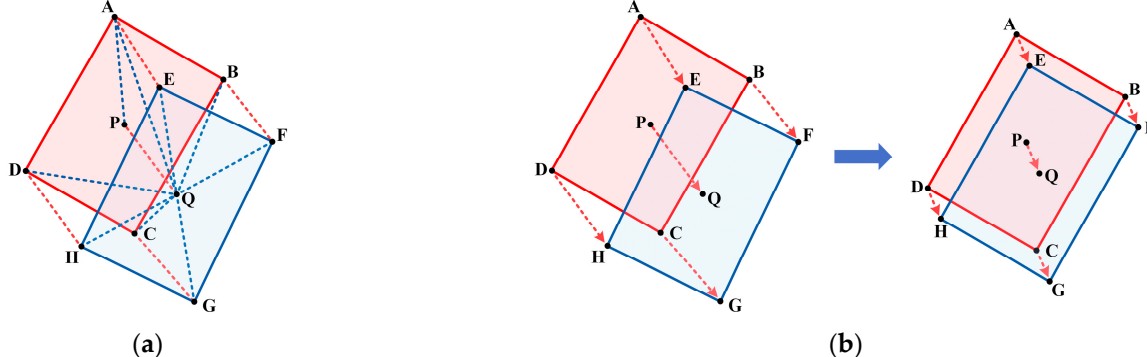

(**a**)                                               (**b**)

**Figure 9.** The schematic diagram of the TDIoU loss. (**a**) The computation of $\mathcal{R}_{TDIoU}$. The red and blue boxes indicate the predicted box $B^{pb}$ and the ground truth $B^{gt}$, respectively. The red and blue lines denote the distance between sampling points; (**b**) the process of bounding box regression guided by TDIoU loss. After back-propagation, the model tends to pull the centroids and vertices of the anchor/proposal toward the corresponding points of the ground truth until they overlap.

Note that each group of corresponding sampling points is exploited to construct independent triangles in $\mathcal{R}_{TDIoU}$. To illustrate this process, here we use the vertices $A$ and $E$. As shown in Figure 9a, we use $A$, $E$, and the centroid of $B^{gt}$, $Q$, to construct $\Delta AEQ$, which obviously satisfies $|AE| < |AQ| + |EQ|$. Then, $|AE|$ is put into the numerator of $\mathcal{R}_{TDIoU}$ to directly measure the distance between the vertices $A$ and $E$, while $|AQ|$ and $|EQ|$ are introduced into the denominator of $\mathcal{R}_{TDIoU}$ as part of the reference term. In this way, we finally establish the entire reference term by traversing each group of sampling points, specifically as follows:

$$
\begin{aligned}
|AE| &< |AQ| + |EQ| \\
|BF| &< |BQ| + |FQ| \\
|CG| &< |CQ| + |GQ| \\
|DH| &< |DQ| + |HQ| \\
|PQ| &< |AP| + |AQ|
\end{aligned}
\tag{15}
$$

In the denominator reference term of $\mathcal{R}_{TDIoU}$, the triangle distance plays a similar role to the diagonal length of the smallest enclosing box, ensuring that the value of the penalty term is limited to $[0, 1)$. The difference is that the computing process of the triangle distance is much simpler than that of the latter as it only involves the computation of the distance between two points, which is able to save more training resources and time.

Overall, our TDIoU loss is a unified solution to all the above requirements. Compared to other bounding box regression losses, it has several advantages in rotation detection, as follows:

1.      The TDIoU loss inherits all the virtues of existing IoU-based losses. As shown in Figure 4c, though the width $w$ and the height $h$ are not directly used in $\mathcal{R}_{TDIoU}$, TDIoU loss can reflect the overall difference between $B^{pb}$ and $B^{gt}$, and is sensitive to aspect ratio changes. As an improvement to CDIoU, the centroid distance is introduced in $\mathcal{R}_{TDIoU}$ to speed up bounding box alignment. As illustrated in Figure 9b, the bounding box regression guided by TDIoU loss tends to pull the centroids and vertices of the anchor/proposal toward the corresponding points of the ground truth until they overlap. This process steadily matches the location, shape, and orientation of $B^{pb}$ and $B^{gt}$;

2.      By measuring the sampling point distance, $\mathcal{R}_{TDIoU}$ realizes the implicit encoding of the relationship between each parameter. As shown in Figure 6, even when $B^{pb}$ and $B^{gt}$ are in a containment relationship, TDIoU loss is able to reflect the angle difference without directly introducing the angle $\theta$, thus, fundamentally immunizing the angular boundary discontinuity. Hence, our TDIoU loss fulfills **Requirements 2 and 3**;

3. The penalty term of TDIoU loss takes into account the computing complexity by using triangles formed by each group of sampling points to construct the denominator, which significantly reduces the training time and satisfies **Requirement 4.**

Additionally, as a novel training metric, TDIoU loss has the following properties:

1. Here, $0 \leq \mathcal{R}_{TDIoU} < 1$. The lower the value of $\mathcal{R}_{TDIoU}$, the higher the similarity between two boxes; the higher the value of $\mathcal{R}_{TDIoU}$, the higher the difference between two boxes.
2. Here, $0 \leq L_{TDIoU} < 2$. When two bounding boxes are completely coincident, $L_{TDIoU} = 0$. When two bounding boxes are far apart, $L_{TDIoU} \to 2$.

### 4.3. Attention-Weighted Feature Pyramid Network

In this part, we introduce the main idea of the proposed attention-weighted feature pyramid network (AW-FPN), which improves the conventional feature fusion networks from the following two aspects: the connection pathway and the fusion method.

#### 4.3.1. Skip-Scale Connections

First used as the identity mapping shortcut in residual blocks [57–59], the skip connection has been a significant component in convolutional networks. In BiFPN, same-level features at different scales are fused via transverse skip-scale connections. However, this single same-level feature reuse neglects the semantic interactions between cross-level features and fails to avoid the semantic loss during layer-to-layer transmission. To search for better network topology, NAS-FPN uses the neural architecture search (NAS) technique. Although it has a haphazard structure that is difficult to interpret, it can guide us in designing a more preferable feature network. As shown in Figure 2c, NAS-FPN contains not only transverse skip-scale connections but also longitudinal skip-scale connections.

Motivated by the above analysis, we devise a more effective feature pyramid network structure, as demonstrated in Figure 10. First, we retain transverse skip-scale connections used in BiFPN for the same-level feature reuse while avoiding adding much cost. Second, to enhance semantic interactions between features of different resolutions, two types of longitudinal skip-scale connections are added in the bi-directional pathways, as follows:

1. **Top-down skip-scale connections**, which integrate higher-level semantic information into lower-level features to improve the classification performance;
2. **Bottom-up skip-scale connections**, which incorporate shallow positioning information into higher-level features to locate small ship targets more accurately.

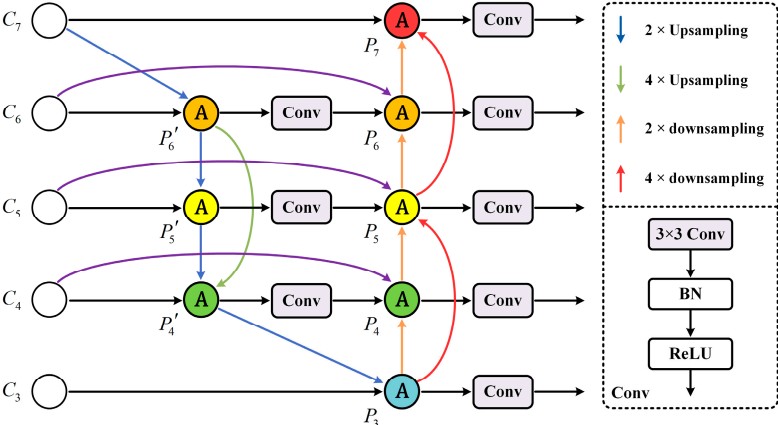

**Figure 10.** Architecture of AW-FPN. Where **A** denotes the attention-weighted feature fusion (AWF).

#### 4.3.2. Attention-Weighted Feature Fusion (AWF)

When fusing features of inconsistent semantics and scales, a common approach is to directly add them together. The BiFPN assigns a learnable scalar weight for each connection pathway. Nevertheless, in the case of considerable variations in target scales, these linear fusion

methods still face obstacles. The AFF [49] provides a non-linear attentional feature fusion scheme. To some extent, our proposed attention-weighted feature fusion (AWF) mechanism can be regarded as its follow-up work, but differs in at least three aspects, as follows:

1. While AFF focuses only on the channel attention, neglecting the spatial context aggregation, our AWF gathers global and local feature contexts in both a multi-scale channel attention module (MCAM) and multi-scale spatial attention module (MSAM);
2. The attentional feature fusion strategy in AFF is restricted to two cross-level features, while our AWF extends it to circumstance of multiple input features;
3. To extract the global channel descriptor, AFF employs only average-pooled features. However, a single average-pooling squeeze may result in the loss of specific spatial information. Hence, to capture finer grained global descriptors, the proposed MCAM and MSAM adopt both average-pooling and max-pooling operations.

Figure 11 describes the process of implementing the AWF. The given $N$ input features from different pyramid levels $F_n \in \mathbb{R}^{C \times H_n \times W_n}$ ($n = 1, 2, \cdots, N$). As they are of different widths and heights, we resize them to the same resolution in advance, as follows:

$$\textit{Resize}: F_n \rightarrow F'_n \in \mathbb{R}^{C \times H \times W} \tag{16}$$

where *Resize* is an upsampling or downsampling operation. To integrate the information flows of different scales from multiple inputs, we first combine them to construct a fully context aware initial integration $U \in \mathbb{R}^{C \times H \times W}$, where $\oplus$ is an element-wise summation, as follows:

$$U = F'_1 \oplus F'_2 \oplus \cdots \oplus F'_n \tag{17}$$

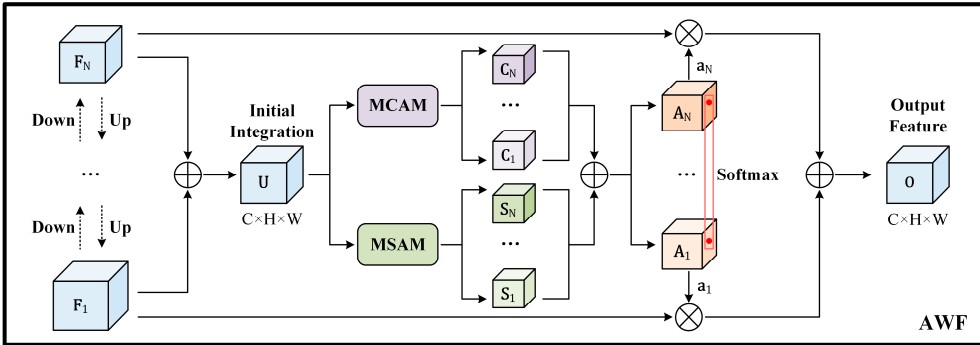

**Figure 11.** Diagram of the AWF. To generate the non-linear fusion weight $a_n$, an element-wise softmax is performed on the integrated attention descriptors $A_n$, fused by multi-scale channel attention $C_n$ and multi-scale spatial attention $S_n$.

Then, to aggregate global and local feature contexts, the initial integration **U** is transmitted to two parallel multi-scale attention modules MCAM and MSAM, as shown in Figure 12.

**MCAM**—to polymerize global spatial information for each channel, we employ both average-pooling and max-pooling operations to squeeze the spatial dimension of **U**, so as to generate two distinct channel-wise statistics. Next, we merge them via an element-wise summation to obtain a refined global channel descriptor. Meanwhile, we follow the idea of AFF to aggregate local channel contexts by altering the pooling size. A simple approach is to directly use **U** as the local channel descriptor. After that, the global and local channel descriptors are fed into two independent excitation branches. As the fully connected layer used in [46,48] cannot be directly performed on the three-dimensional tensor, we adopt the convolution operation with a kernel size of $1 \times 1$, which only uses point-wise channel interactions at each spatial location to learn the non-linear association between channels.

The global channel context $C_g(U)$ and the local channel context $C_L(U)$ are defined as follows:

$$C_g(U) = \mathcal{B}\left(Conv_2^{1\times 1}\left(\delta\left(\mathcal{B}\left(Conv_1^{1\times 1}(AvgPool(U) \oplus MaxPool(U))\right)\right)\right)\right) \quad (18)$$

$$C_L(U) = \mathcal{B}\left(Conv_2^{1\times 1}\left(\delta\left(\mathcal{B}\left(Conv_1^{1\times 1}(U)\right)\right)\right)\right) \quad (19)$$

where $C_g(U) \in \mathbb{R}^{NC\times 1\times 1}$ and $C_L(U) \in \mathbb{R}^{NC\times H\times W}$. Here, $\mathcal{B}$ denotes the batch normalization [60]. Additionally, $\delta$ is the ReLU function, and $Conv^{1\times 1}$ is the $1 \times 1$ convolution. To simplify computation, the first convolution of each branch is used for channel reduction, while the second is used to restore the channel dimension. Hence, the numbers of filters of $Conv_1^{1\times 1}$ and $Conv_2^{1\times 1}$ are set to $C/r$ and $NC$, where $r$ is the channel reduction ratio. Then, $C_g(U)$ and $C_l(U)$ are fused via the broadcasting mechanism to construct the multi-scale channel context $C(U)$. This can be seen as follows:

$$C(U) = C_g(U) \oplus C_l(U) \quad (20)$$

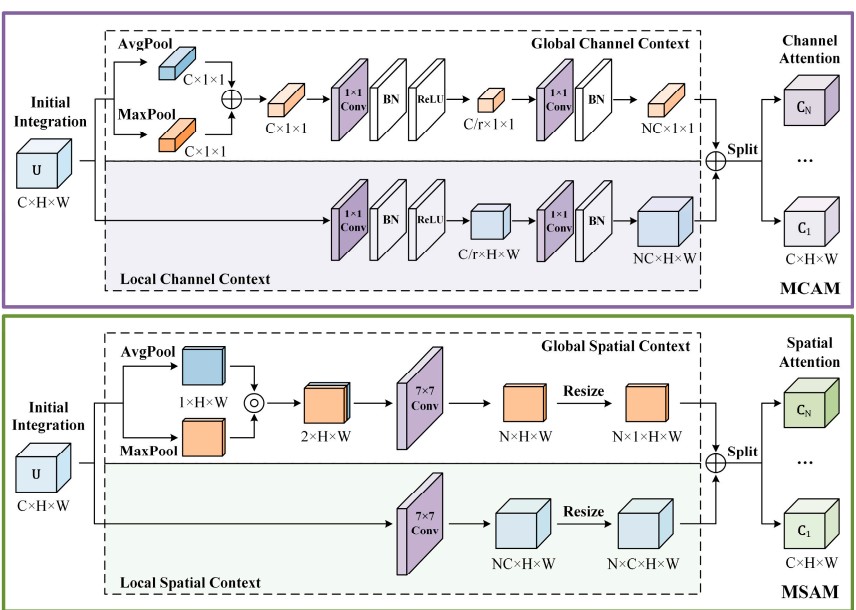

**Figure 12.** Diagram of each attention sub-module. Multi-scale feature contexts are aggregated in both MCAM and MSAM.

Since $C(U) \in \mathbb{R}^{NC\times H\times W}$ is a channel context aggregation of $N$ input features, it is subsequently split into $C_n \in \mathbb{R}^{C\times H\times W}$ as the multi-scale channel attention for each input.

**MSAM**—similarly to MCAM, to learn the global and local cross-spatial relationships of the initial integration **U,** we use two parallel branches. First, to obtain a refined global spatial descriptor, we perform the average-pooling and max-pooling operations along the channel dimension and concatenate them, while the initial integration is simply treated as the local spatial descriptor. Then, the convolution layer with a kernel size of $7 \times 7$, which has a broader receptive field, is selected as the spatial context aggregator to encode emphasized or suppressed positions of spatial descriptors. On this basis, the global spatial context $S_g(U)$ and the local spatial context $S_l(U)$ can be defined as follows:

$$S_g(U) = Resize\left(Conv_1^{7\times 7}(AvgPool(U) \odot MaxPool(U))\right) \quad (21)$$

$$S_l(U) = Resize\left(Conv_2^{7\times 7}(U)\right) \quad (22)$$

where $S_g(U) \in \mathbb{R}^{n \times 1 \times H \times W}$ and $S_l(U) \in \mathbb{R}^{n \times C \times H \times W}$. Here, $\odot$ indicates a concatenate operation. The numbers of filters of $Conv_1^{7 \times 7}$ and $Conv_2^{7 \times 7}$ are set to $N$ and $NC$. As the convolution outputs of the two branches cannot be added directly, we resize them and then fuse them via the broadcast mechanism to obtain the multi-scale spatial context $S(U) \in \mathbb{R}^{n \times C \times H \times W}$.

$$S(U) = S_g(U) \oplus S_l(U) \tag{23}$$

We split $S(U)$ into $S_n \in \mathbb{R}^{C \times H \times W}$ as the multi-scale spatial attention, and the integrated attention descriptor $A_n \in \mathbb{R}^{C \times H \times W}$ can be computed by $A_n = C_n \oplus S_n$. Next, to generate the non-linear fusion weight $a_n$ for each input feature, a softmax operation is executed on each group of corresponding elements of all attention descriptors $A_n$.

$$a_n = \frac{e^{A_n}}{e^{A_1} \oplus e^{A_2} \oplus \cdots \oplus e^{A_n}} \tag{24}$$

Each element $a_n^{x, y, z}$ of $a_n$ is a real number between 0 and 1 and fulfills $\sum_{n=1}^{n} a_n^{x, y, z} = 1$. As $a_n \in \mathbb{R}^{C \times H \times W}$ have the same size as resized features $F_n'$, they preserve and emphasize the subtle details in all inputs, enabling high-quality soft feature selections between $F_n'$.

$$O = (a_1 \otimes F_1') \oplus (a_2 \otimes F_2') \oplus \cdots \oplus (a_n \otimes F_n') \tag{25}$$

Here, **O** signifies the final fused feature and $\otimes$ implies an element-wise multiplication.

### 4.3.3. The Forward Process of the AW-FPN

Our ultimate AW-FPN combines both multiple skip-scale connections and attention-weighted feature fusion. As shown in Figure 10, it takes level 3–7 features extracted by the backbone network as the input $C^{in} = \{C_3, C_4, C_5, C_6, C_7\}$, where $C_i$ denotes a feature level with a resolution of $1/2^i$ of the input image. The top-down and bottom-up aggregation pathways are constructed successively. Here, we take level 5 as an example to illustrate the forward process. On the top-down pathway, the intermediate feature of level 6 ($P_6'$) is upsampled $2 \times$ and then fused with $C_5$ via AWF, followed by a $3 \times 3$ convolution to generate the intermediate feature $P_5'$. On the bottom-up pathway, the outputs of levels 3 and 4 ($P_4$ and $P_3$) are subjected to $4 \times$ and $2 \times$ downsampling operations, respectively, and then fused with $C_5$ and $P_5'$. The final output $P_5$ is generated by the $3 \times 3$ convolution, as follows:

$$P_5' = Conv\big(\text{AWF}\big(C_5, \, Resize(P_6')\big)\big) \tag{26}$$

$$P_5 = Conv\big(\text{AWF}\big(C_5, \, P_5', \, Resize(P_4), \, Resize(P_3)\big)\big) \tag{27}$$

where *Conv* implies the $3 \times 3$ convolution, which is followed by a batch normalization operation and a ReLU function. All other feature levels are constructed in a similar way.

## 5. Rotated-SARShip Dataset

In this section, we introduce the collection process and data statistics of our proposed rotated-SARShip dataset (RSSD) for arbitrary-oriented ship detection in SAR images.

### 5.1. Original SAR Image Acquisition

Table 2 provides detailed information of the original SAR imageries used to establish our RSSD. First, from the Copernicus Open Access Hub [61], we downloaded three raw Sentinel-1 images with a resolution of 5 m × 20 m, characterized by large scales and wide coverages (25,340 × 17,634 pixels on average). As shown in Figure 13, the imagery acquisition areas are selected in the Dalian Port, Panama Canal, and the Tokyo Port (these ports have huge cargo throughputs, and the canal has busy trade). In general, the polarization, imaging mode, and the incident angle of sensors tend to influence the imaging condition of SAR images to a certain extent. For the Sentinel-1 images, the basic polarimetric combination is VV and VH. The imaging mode is interferometric wide swath (IW), which is

the primary sensor mode for data acquisition in marine surveillance zones. Furthermore, to minimize redundant interferences, such as foreshortening, layover, and shadowing of vessels, we choose an incident angle of 27.6~34.8° [32].

**Table 2.** Detailed information of the original SAR imageries used to establish our RSSD.

| No. | Data Source | Polarization | Imaging Mode | Incident Angle (°) | Resolution (m) | Image Size (Pixel) | Location | Date and Time |
|---|---|---|---|---|---|---|---|---|
| 1 | Sentinel-1 | VV, VH | IW | 27.6~34.8 | 5 × 20 | 25,313 × 16,704 | Dalian Port | 5 October 2021, 09:48:20 |
| 2 | Sentinel-1 | VV, VH | IW | 27.6~34.8 | 5 × 20 | 25,136 × 19,488 | Panama Canal | 30 September 2021, 11:06:41 |
| 3 | Sentinel-1 | VV, VH | IW | 27.6~34.8 | 5 × 20 | 25,480 × 16,709 | Tokyo Port | 1 October 2021, 08:41:23 |
| 4~256 | Gaofen-3 (AIR-SARShip) | Single, VV | SpotLight, SM | – | 1, 3 | 1000 × 1000, 3000 × 3000 | – | – |
| 257~5792 | Sentinel-1, TerraSAR-X (HRSID) | HH, HV, VV | S3-SM, ST, HS | 27.6~34.8, 20~45, 20~60, 20~55 | 0.5, 1, 3 | 800 × 800 | Barcelona, Sao Paulo, Houston, etc. | – |

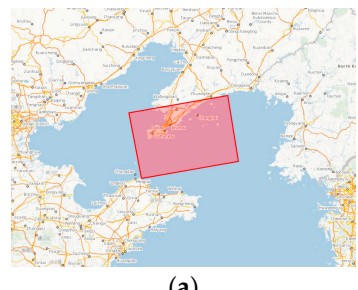 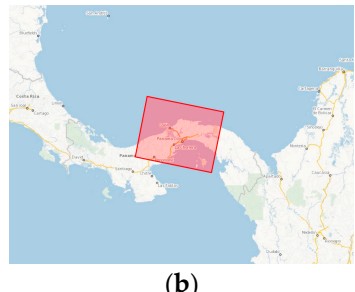 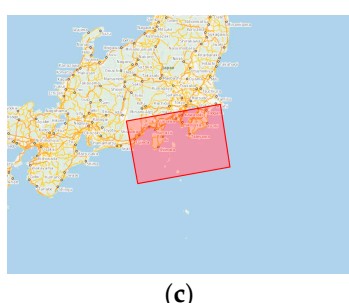

(**a**) (**b**) (**c**)

**Figure 13.** Coverage areas of No. 1–3 Sentinel-1 images. (**a**) The Dalian Port; (**b**) the Panama Canal; (**c**) the Tokyo Port.

To ensure complex and diverse image scenes, we also screen 252 and 5535 SAR images from AIR-SARShip [31] and HRSID [32], respectively. As shown in Table 2, the HRSID images shot by Sentinel-1 and TerraSAR-X have resolutions of 0.5 m, 1 m, and 3 m. The polarizations are HH, HV, and VV, and the imaging modes are S3-StripMap (S3-SM), Staring SpotLight (ST), and High-Resolution SpotLight (HS). The AIR-SARShip images from Gaofen-3 have resolutions of 1 m and 3 m, polarizations of single and VV, and imaging modes of SpotLight and StripMap (SM). Since these images have different resolutions and imaging conditions, ships in them usually appear in different forms. Notably, images with a resolution of less than 3 m can retain the detailed features of ships, while images with a resolution of 5 m × 20 m can increase the number of small ship targets.

*5.2. SAR Image Pre-Processing and Splitting*

The above original SAR imageries still need to be pre-processed before annotation. To display recognizable features of ships, we first apply the Sentinel-1 toolbox [62] to convert the raw Sentinel-1 data into grayscale images in the 16-bit tag image file format (TIFF), followed by geometrical rectification and radiometric calibration operations. Since images selected from AIR-SARShip and HRSID have already undergone the above processing, we directly perform the de-speckling operation on all the original images to suppress the influence of background noise. Finally, we transform all images into portable network graphics (PNG) files in the same format as the DOTA dataset [63].

Due to the side-scan imaging mechanism of SAR satellites, the original SAR imagery generally has a huge size and should be split into ship slices to fit the input size of CNN-based detectors. First, to avoid duplicate splitting, the offshore areas with a relatively dense ship distribution are separated from the images in advance [32]. After that, a sliding window of 800 × 800 pixels is used to shift over the whole image with a stride of 600 pixels in width

and height (25% overlap rate) to preserve the relatively intact features of ships. Since the images screened from HRSID have already been cropped to the expected size matching the network input, the splitting operation is performed only on the images from Sentinel-1 and AIR-SARShip. Furthermore, we reserve the complex inshore scenes containing ships and artificial facilities and remove the negative samples with only pure background.

*5.3. Dataset Annotation*

With the assistance of the official document and the Sentinel-1 toolbox, we can easily acquire the precise imaging time and geographic location of each Sentinel-1 image, which will help the automatic identification system (AIS) and Google Earth to provide support for the annotation work. As shown in Figure 14, we first identify the approximate location of the imaging area of each Sentinel-1 image in AIS and Google Earth. Since AIS provides the movement trajectories of most ships around the time the images were shot, it is possible to grasp the approximate distribution of ships and estimate their possible positions in the imaging area. Subsequently, we match the AIS message with each Sentinel-1 image and determine the topographical features and marine conditions of the coverage area with the help of Google Earth. On this basis, we adopt RoLabelImg [64] to annotate the oriented bounding boxes of ships, obtaining relatively accurate ground truths. To ensure that the annotations meet the requirements of most rotation detectors, we convert them to the DOTA format, using four ordered vertices to represent ship ground truths, as shown in Figure 15. In fact, there are still some islands and reefs incorrectly labeled as ships. Thus, we employ Google Earth for further in-depth inspection and correction to ensure the accuracy of the annotations. Note that since the specific shooting information of the images from AIR-SARShip and HRSID cannot be acquired directly, we first refer to their original horizontal ground truths and carefully check whether there are errors and omissions. Then, we annotate them with more elaborate oriented bounding boxes.

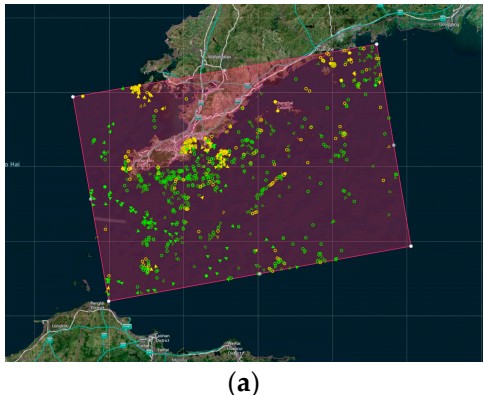 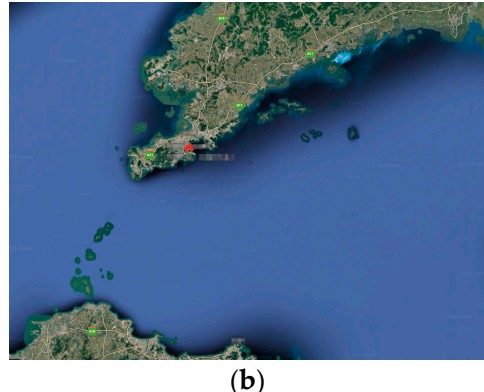

(**a**)       (**b**)

**Figure 14.** AIS query and Google Earth correction (take No.1 Sentinel-1 image as an example). (**a**) AIS information of the coverage. Marks of different shapes and colors represent different types of ships; (**b**) corresponding Google Earth image.

So far, we have established the RSSD, and there have been 8013 SAR images with corresponding annotation files, including 21,479 ship targets annotated by rotated ground truths. Figure 16 displays ship ground truth annotations of diverse SAR images in RSSD.

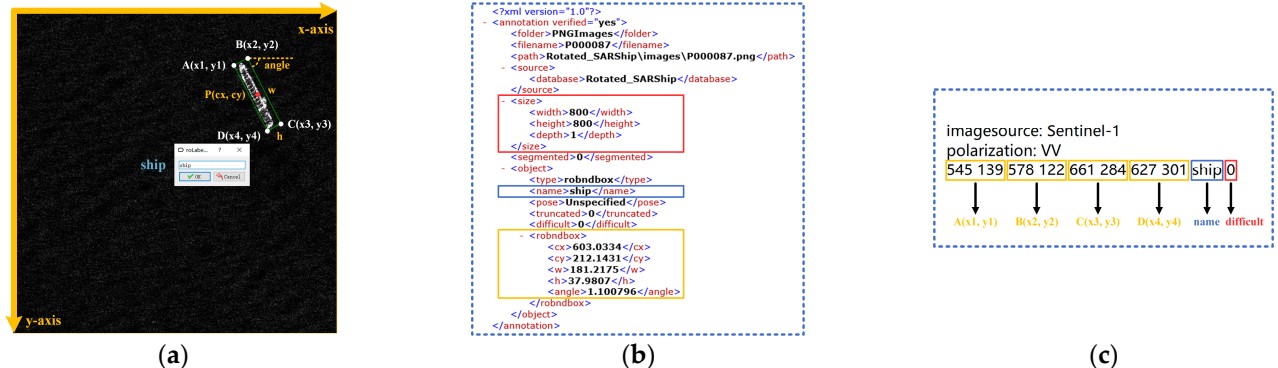

(a) (b) (c)

**Figure 15.** The ship annotation. (**a**) The OBB label in a SAR image; (**b**) the xml label file annotated by RoLabelImg. Each ship target is represented by an oriented bounding box as its ground truth. Where (*cx*, *cy*) is the centroid, *w* and *h* denote the width and height, and *θ* indicates the rotation angle; (**c**) the txt label file in DOTA format. Each ship is represented by four ordered vertices. Note that the top-left vertex is taken as the starting point, and the four vertices are arranged in clockwise order.

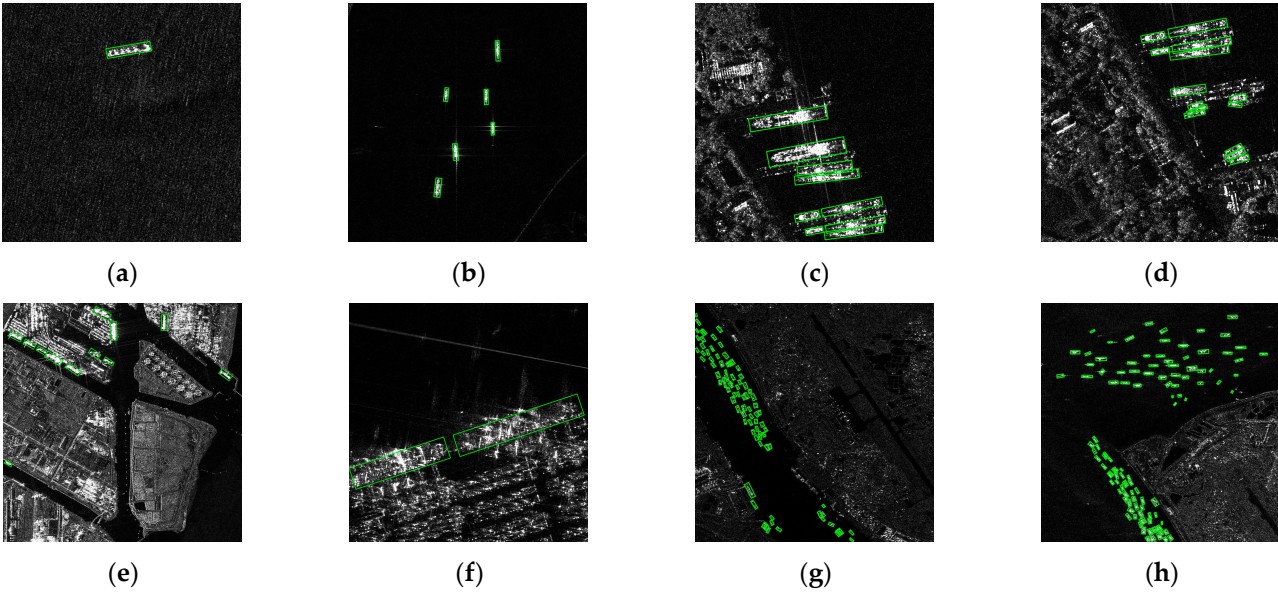

(a) (b) (c) (d)

(e) (f) (g) (h)

**Figure 16.** Ship ground truth annotations of diverse SAR images in RSSD. Real ships are accurately marked in green OBBs. (**a**) Offshore single ship; (**b**) offshore multiple ships; (**c**,**d**) densely arranged ships; (**e**) ships lying off the port; (**f**) ships with large aspect ratios; (**g**,**h**) small ships in the canal.

*5.4. Statistical Analysis on the RSSD*

Figure 17 visualizes the comprehensive statistical comparison between our RSSD and SSDD, both of which adopt OBB annotations. As Table 3 shows, 70% of the RSSD images are randomly selected as the training set, and 30% are selected as the test set. For the SSDD, we divide all images in the ratio of 8:2 according to [28]. As shown in Table 1, SSDD contains 1160 SAR images with 2587 annotated ships, indicating that each image contains only 2.2 ships on average, while in our dataset, each image contains about 2.7 ships. Figure 17a,e display the width and height distribution of ship ground truths. Compared to the extreme funnel-like distribution of SSDD, our RSSD features a more uniform ship size distribution and more prominent multi-scale characteristics. As per Figure 17b,f, the aspect ratio of ship ground truths in the SSDD is generally below 3, whereas it is concentrated in the range of 2~5 in the RSSD, indicating that most instances in our dataset are with relatively high aspect ratios. Since the difficulty in detecting ships typically increases with the aspect ratio, our RSSD is more challenging compared to other datasets. As per Figure 17c,g, according to the MS COCO evaluation metric [65], the numbers of small ships ($Area_{OBBs} < 1024$ pixels),

medium ships ($1024 < Area_{\text{OBBs}} < 9216$ pixels), and large ships ($Area_{\text{OBBs}} > 9216$ pixels) in the RSSD are 13,369, 7741, and 369, respectively, (62.24%, 36.04%, and 1.72% of all ships, respectively), while in the SSDD, the proportions are 71.12%, 28.30%, and 0.58%, respectively. Ships in both datasets are relatively small in size but have large variations in scale. As shown in Figure 17d,h, the angle distribution of ship ground truths in the RSSD is more balanced than that in the SSDD. This ensures that rotation detectors learn the multi-angle features better.

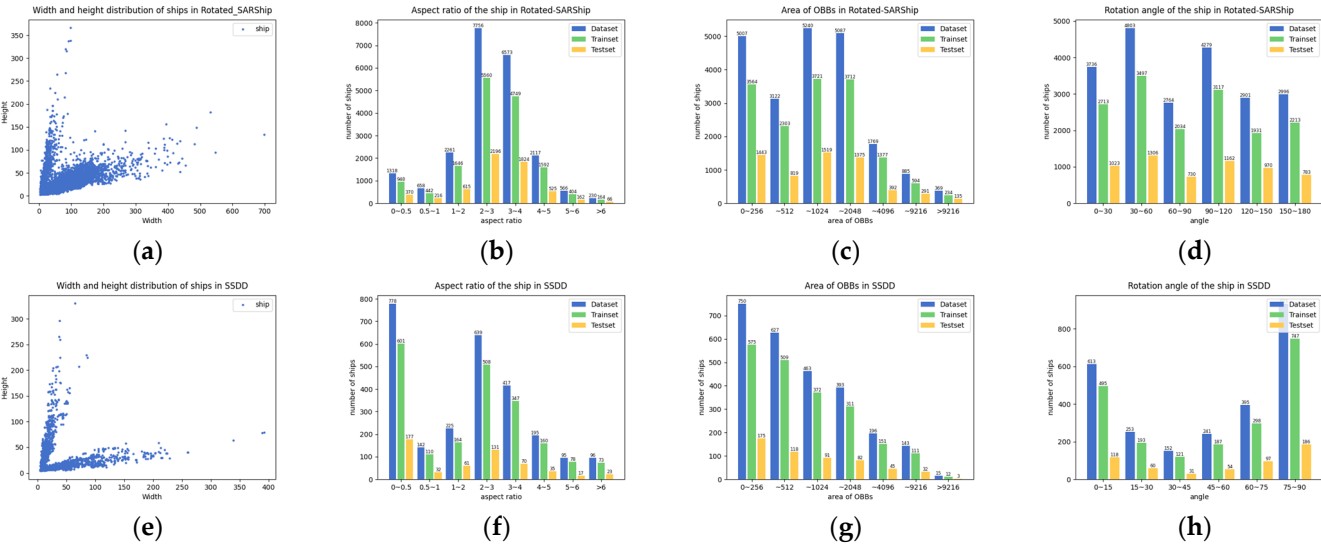

**Figure 17.** Statistical comparison between the proposed RSSD and the SSDD. Here, (**a**) and (**e**) show the width and height distribution of ship ground truths in RSSD and SSDD, respectively; (**b**) and (**f**) display the aspect ratio distribution of ship OBBs; (**c**) and (**g**) indicate the area distribution of ship OBBs; (**d**) and (**h**) show the rotation angle distribution of ship OBBs.

**Table 3.** Details of dataset division.

| Dataset | Train | Test | All | Inshore (Test) | Offshore (Test) |
|---|---|---|---|---|---|
| RSSD (ours) | 5692 | 2321 | 8013 | 479 | 1842 |
| SSDD | 928 | 232 | 1160 | 46 | 186 |
| HRSC2016 | 617 | 444 | 1061 | – | – |

Based on the above analysis, it is obvious that the ship targets in our RSSD not only differ significantly in orientation degrees but also have multi-scale characteristics, which provides a challenging benchmark for arbitrary-oriented ship detection in SAR images.

## 6. Experiments and Discussion

In this section, we first present the benchmark datasets, implementation details, and evaluation metrics. Then, extensive comparative experiments with existing methods are carried out to verify the superiority and robustness of our approach. Meanwhile, comprehensive discussions are provided to analyze and interpret the experimental results.

### 6.1. Benchmark Datasets and Implementation Details

The proposed rotated-SARShip dataset (RSSD) and the public SAR ship detection dataset (SSDD), specific information about which is provided in Section 5, are used to evaluate the performance of the proposed method. In our experiments, all SSDD images are resized to $512 \times 512$ pixels, with padding operation to avoid distortion, while the RSSD images of $800 \times 800$ pixels are directly used as the network input. The ratio of training set to test set for the RSSD is set to 7:3, while that for the SSDD is set to 8:2. To better assess the

performance of our approach in different SAR scenes, we further divide the test sets into inshore and offshore scenes. Figure 18 and Table 3 show the details of dataset division.

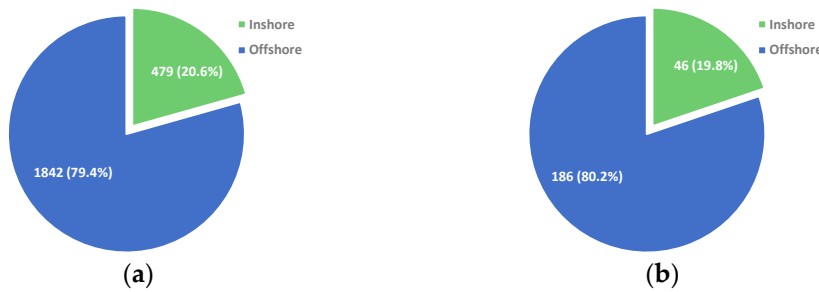

**Figure 18.** The proportion of inshore and offshore scenes in the test sets of (**a**) RSSD and (**b**) SSDD.

Furthermore, a public benchmark for OBB-based ship detection in optical remote sensing images, the HRSC2016 dataset [66], is used to verify the generalization ability of the proposed method across different scenarios. It contains 1061 high-resolution aerial images, including 2976 different types of ships annotated by oriented bounding boxes, with the image size ranging from $300 \times 300$ to $1500 \times 900$ pixels. We employ the training (436 images) and validation (181 images) sets for training, and the test set (444 images) for testing. All images are resized to $800 \times 512$ pixels without altering the original aspect ratio.

The experiments are conducted on the platform with Ubuntu 18.04 OS, 32 GB of RAM, and a NVIDIA GTX 1080Ti GPU. For all datasets, we train the models in 72 epochs. The SGD optimizer is adopted with a batch size of 2 and an initial learning rate of 0.0025. The momentum and weight decay are 0.9 and 0.0001, respectively. As for the learning schedule, we apply the warmup strategy for 500 iterations, and the learning rate is dropped 10-fold at each decay step. If not specified, ResNet50 [57] is employed as the default backbone network. Its parameters are initialized by ImageNet pretrained weights. For fair comparisons with other methods and to avoid over-fitting, we only use random flipping and rotation for data augmentation in the training phase. If not specified, no extra tricks are used.

### 6.2. Evaluation Metrics

To qualitatively and quantitatively evaluate the detection performance of different methods in our experiments, two normative metrics, the precision–recall curve (P–R curve) and average precision (AP), are leveraged. Specifically, the precision and the recall can be expressed as follows:

$$Precision = \frac{TP}{TP + FP} \tag{28}$$

$$Recall = \frac{TP}{TP + FN} \tag{29}$$

where $TP$ (true positives), $FP$ (false positives), and $FN$ (false negatives) represent the number of correctly detected ships, false alarms, and undetected ships, respectively. The P–R curve, with precision as the $y$-axis and recall as the $x$-axis, reveals the relationship between these two metrics. The AP is defined as the area under the P–R curve, as follows:

$$AP = \int_0^1 P(R)dR \tag{30}$$

where $P$ and $R$ indicate the precision and recall, respectively. The AP evaluates the overall performance of detectors under different IoU thresholds (0.5 by default) and, the larger the value, the better the performance. Furthermore, we use the total training time as a metric to evaluate the computing complexity and training efficiency of different losses.

### 6.3. Ablation Study

In this part, we first introduce two robust rotation detectors as baselines. On this basis, a series of component-wise experiments on the RSSD, the SSDD, and the HRSC2016 are carried out to validate the effectiveness of the proposed TDIoU loss and AW-FPN.

#### 6.3.1. Baseline Rotation Detectors

Two rotation detectors, RetinaNet [17] and CS$^2$A-Net [67], are selected as baselines in our experiments. As a typical single-stage detector, RetinaNet consists of a backbone network, a feature pyramid network, and detection heads. It uses a ResNet [57] to generate a multi-scale feature pyramid and attaches a detection head to each pyramid level ($P_3$ to $P_7$). Each detection head is made up of a classification sub-network and a regression sub-network. To implement a RetinaNet-based rotation detector (RetinaNet-R), we modify the regression output to an OBB ($cx$, $cy$, $w$, $h$, and $\theta$) under the long-edge definition, where ($cx$, $cy$), $w$, $h$, and $\theta$ denote the centroid, the width, the height, and the angle, respectively, and $\theta \in [-45°, 135°)$. Accordingly, the angle $\theta$ is taken into consideration in the anchor generation. At each pyramid level, we set anchors in three aspect ratios, {1:2, 1:1, and 2:1}, three scales, {1, $2^{1/3}$, and $2^{2/3}$}, and six angles, {$-45°$, $-15°$, $15°$, $45°$, $75°$, and $105°$}. The proposed TDIoU loss and AW-FPN can be easily embedded into RetinaNet-R, as shown in Figure 19a.

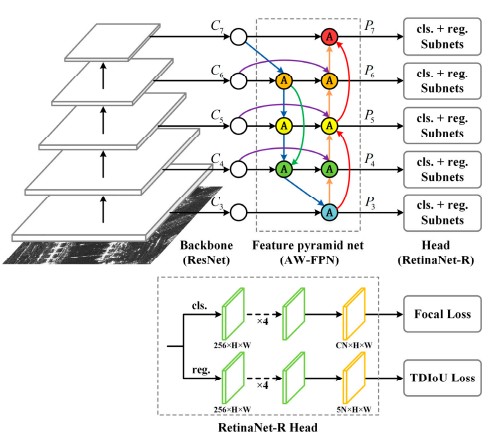
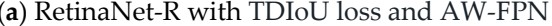

(**a**) RetinaNet-R with TDIoU loss and AW-FPN

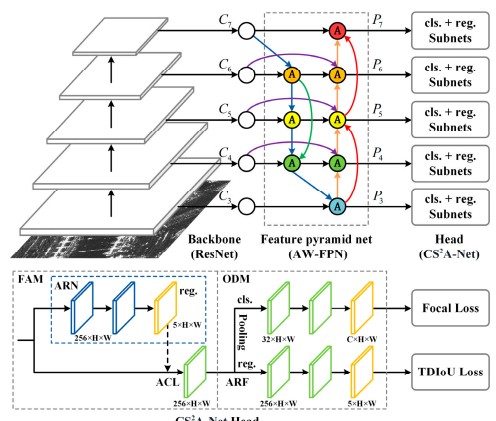

(**b**) CS$^2$A-Net with TDIoU loss and AW-FPN

**Figure 19.** Architectures of two baselines. As a plug-and-play scheme, TDIoU loss and AW-FPN can be easily embedded into the above rotation detectors. (**a**) The regression output of RetinaNet is modified to an OBB under long-edge definition. Here, 'C' denotes the number of categories, and 'N' denotes the number of anchors on each feature point; (**b**) the CS$^2$A-Net head consisting of the FAM and ODM can be cascaded to improve accuracy. The number of cascade heads is set to 2 by default.

The CS$^2$A-Net is an advanced rotation detector based on the RetinaNet architecture. Its detection head consists of a feature alignment module (FAM) and an oriented detection module (ODM), which can be cascaded to improve accuracy. The FAM uses an anchor refinement network (ARN) to generate refined rotated anchors, and then sends refined anchors and input features to an alignment convolution layer (ACL) to learn aligned features. In ODM, the active rotating filter (ARF) learns orientation-sensitive features, and then a pooling operation extracts the orientation-invariant features for classification and regression. Our TDIoU loss and AW-FPN can also be integrated into CS$^2$A-Net, as shown in Figure 19b.

The multi-task loss function of two baseline detectors is defined as follows:

$$L = \frac{1}{N} \sum_{N=1}^{n} S_n L_{\text{reg}} \left( B_n^{pb}, B_n^{gt} \right) + \frac{\lambda_2}{N} \sum_{N=1}^{n} L_{\text{cls}} \left( p_n^{pb}, p_n^{gt} \right) \qquad (31)$$

where $\v{}_1$ and $\v{}_2$ indicate the loss balance hyper-parameter and are set to 1 by default, $N$ denotes the number of anchors in a mini-batch, and $S_n$ is a binary value ($S_n = 1$ for positive anchors and $S_n = 0$ for negative anchors). The vectors $B_n^{pb}$ and $B_n^{gt}$ denote the locations of the $n$-th predicted box and the corresponding ground truth, respectively. The values $p_n^{pb}$ and $p_n^{gt}$ indicate the predicted classification score and the true label of the $n$th object, respectively. In our experiments, the regression loss $L_{reg}$ is set as the smooth L1 loss, the TDIoU loss, etc. The classification loss $L_{cls}$ is set as the focal loss [17], as follows:

$$L_{focal}(p_t) = -\text{ff}_t(1 - p_t)^{\text{fl}}\log(p_t) \tag{32}$$

where $(1 - p_t)^\gamma$ and $\alpha_t$ are two modulation factors that satisfy the following conditions:

$$p_t = \begin{cases} p_n^{pb}, & p_n^{gt} = 1 \\ 1 - p_n^{pb}, & \text{otherwise} \end{cases} \text{ and } \alpha_t = \begin{cases} \alpha, & p_n^{gt} = 1 \\ 1 - \alpha, & \text{otherwise} \end{cases} \tag{33}$$

where $\alpha$ and $\gamma$ are two hyper-parameters, which are set to 0.25 and 2, respectively, by default.

6.3.2. Effectiveness of the TDIoU Loss

We evaluate the TDIoU loss with two baseline detectors on three datasets, as shown in Tables 4–6. Both detectors adopt ResNet50 and the original FPN. To ensure the objectivity and richness of the ablation study, we implement two approximate IoU losses (IoU-smooth L1 and GWD loss) and five IoU-based losses (IoU, GIoU, CIoU, EIoU, and CDIoU loss) to compare the performance of different regression losses. Only the regression loss is modified, and all other settings remain intact for fair comparisons.

**Table 4.** Comparison of different regression losses on RSSD. Here, R-50-FPN denotes ResNet50 with FPN, **LMI** and **ABD** denote the loss-metric inconsistency and angular boundary discontinuity, respectively, and ✓ indicates that the method has corresponding issue. **Training** represents the total training time (in hours) for 72 epochs with a single GPU and a batch size of 2. Bold items are the best result of each column.

| Detector | Regression Loss | LMI | ABD | Inshore AP | Offshore AP | Test AP | Training (h) |
|---|---|---|---|---|---|---|---|
| RetinaNet-R (R-50-FPN) | Smooth L1 (baseline) | ✓ | ✓ | 44.30 | 91.38 | 72.13 | **10.2** |
| | IoU-smooth L1 [41] | ✓ | | 45.49 (+1.19) | 92.37 (+0.99) | 73.22 (+1.09) | 12.9 |
| | GWD [23] | ✓ | | 48.28 (+3.98) | 93.36 (+1.98) | 74.92 (+2.79) | 12.1 |
| | IoU [36] | | | 47.37 (+3.07) | 93.11 (+1.73) | 74.36 (+2.23) | 12.6 |
| | GIoU [37] | | | 47.43 (+3.13) | 93.16 (+1.78) | 74.43 (+2.30) | 26.2 |
| | CIoU [38] | | | 47.76 (+3.46) | 93.25 (+1.87) | 74.65 (+2.52) | 26.8 |
| | EIoU [39] | | | 48.01 (+3.71) | 93.30 (+1.92) | 74.77 (+2.64) | 26.6 |
| | CDIoU [40] | | | 48.54 (+4.24) | 93.46 (+2.08) | 75.05 (+2.92) | 26.5 |
| | AIoU | | ✓ | NAN | NAN | NAN | – |
| | TDIoU | | | **49.68 (+5.38)** | **94.09 (+2.71)** | **75.93 (+3.80)** | 13.0 |
| CS$^2$A-Net (R-50-FPN) | Smooth L1 (baseline) | ✓ | ✓ | 70.99 | 96.13 | 85.95 | **11.7** |
| | TDIoU | | | **75.17 (+4.18)** | **96.56 (+0.43)** | **87.65 (+1.70)** | 14.8 |

Table 4 shows results on our RSSD. Compared with smooth L1, RetinaNet-R based on approximate IoU losses improves the AP of inshore scenes, offshore scenes, and the entire test set by 1.19~3.98%, 0.99~1.98%, and 1.09~2.79%, respectively. Conventional IoU-based losses improve the AP by 3.07~4.24%, 1.73~2.08%, and 2.23~2.92%, respectively. The proposed TDIoU loss improves the AP by 5.38%, 2.71%, and 3.80%, respectively. Even with the advanced CS$^2$A-Net, TDIoU loss still improves the inshore AP, offshore AP, and test AP by 4.18%, 0.43%, and 1.70%, respectively, indicating that our method dramatically improves ship detection performance, especially in the complex inshore scenes. Similar experimental conclusions are also reflected in the other two datasets. Table 5 shows results on the SSDD. The TDIoU-based RetinaNet-R is improved by 8.50%, 1.01%, and 3.42% on inshore AP,

offshore AP, and test AP, respectively, compared to the approximate IoU losses (2.24~5.88%, 0.27~0.61%, and 0.90~2.25%) and the traditional IoU-based losses (2.94~6.75%, 0.39~0.67%, and 1.17~2.61%). When CS$^2$A-Net is used as the base detector, our TDIoU loss further improves the AP by 3.47%, 0.73%, and 1.51%. Table 6 shows results on the HRSC2016. The RetinaNet-R achieves the best accuracy by using the TDIoU loss (i.e., improvement by 3.49% and 4.39% in terms of the 2007 and 2012 evaluation metrics, respectively). Similarly, our TDIoU loss achieves considerable improvement on CS$^2$A-Net, with an increase of 0.32% and 2.48%, respectively.

**Table 5.** Comparison of different regression losses on SSDD.

| Detector | Regression Loss | LMI | ABD | Inshore AP | Offshore AP | Test AP | Training (h) |
|---|---|---|---|---|---|---|---|
| RetinaNet-R (R-50-FPN) | Smooth L1 (baseline) | ✓ | ✓ | 59.35 | 97.10 | 86.14 | 1.1 |
| | IoU-Smooth L1 [41] | ✓ | | 61.59 (+2.24) | 97.37 (+0.27) | 87.04 (+0.90) | 1.5 |
| | GWD [23] | ✓ | | 65.23 (+5.88) | 97.71 (+0.61) | 88.39 (+2.25) | 1.3 |
| | IoU [36] | | | 62.29 (+2.94) | 97.49 (+0.39) | 87.31 (+1.17) | 1.4 |
| | GIoU [37] | | | 63.36 (+4.01) | 97.57 (+0.47) | 87.68 (+1.54) | 2.9 |
| | CIoU [38] | | | 64.13 (+4.78) | 97.63 (+0.53) | 87.98 (+1.84) | 3.1 |
| | EIoU [39] | | | 64.24 (+4.89) | 97.65 (+0.55) | 88.02 (+1.88) | 3.0 |
| | CDIoU [40] | | | 66.10 (+6.75) | 97.77 (+0.67) | 88.75 (+2.61) | 2.9 |
| | TDIoU | | ✓ | **67.85 (+8.50)** | **98.11 (+1.01)** | **89.56 (+3.42)** | 1.6 |
| CS$^2$A-Net (R-50-FPN) | Smooth L1 (baseline) | | | 75.79 | 98.79 | 92.08 | 1.2 |
| | TDIoU | ✓ | ✓ | **79.26 (+3.47)** | **99.52 (+0.73)** | **93.59 (+1.51)** | 1.7 |

**Table 6.** Comparison of different regression losses on HRSC2016. Here, AP$_{07}$ and AP$_{12}$ indicate the PASCAL VOC 2007 and 2012 metrics, respectively.

| Detector | Regression Loss | LMI | ABD | Test AP$_{07}$ | Test AP$_{12}$ | Training (h) |
|---|---|---|---|---|---|---|
| RetinaNet-R (R-50-FPN) | Smooth L1 (baseline) | ✓ | ✓ | 81.63 | 84.82 | **1.1** |
| | IoU-Smooth L1 [41] | ✓ | | 82.64 (+1.01) | 85.84 (+1.02) | 1.4 |
| | GWD [23] | ✓ | | 83.94 (+2.31) | 87.78 (+2.96) | 1.2 |
| | IoU [36] | | | 83.07 (+1.44) | 86.64 (+1.82) | 1.3 |
| | GIoU [37] | | | 83.22 (+1.59) | 86.83 (+2.01) | 2.8 |
| | CIoU [38] | | | 83.62 (+1.99) | 87.33 (+2.51) | 3.1 |
| | EIoU [39] | | | 83.78 (+2.15) | 87.55 (+2.73) | 3.0 |
| | CDIoU [40] | | | 84.13 (+2.50) | 88.06 (+3.24) | 2.9 |
| | TDIoU | | ✓ | **85.12 (+3.49)** | **89.21 (+4.39)** | 1.5 |
| CS$^2$A-Net (R-50-FPN) | Smooth L1 (baseline) | | | 89.94 | 94.91 | **1.1** |
| | TDIoU | ✓ | ✓ | **90.26 (+0.32)** | **97.39 (+2.48)** | 1.6 |

Figure 20 shows P–R curves of RetinaNet-R using different regression losses on three datasets. The area under the P–R curve of TDIoU loss is always larger than that of the other losses, indicating that the overall performance of our method is better. The possible causes are summarized as follows: (1) Compared to the approximate IoU losses, we fundamentally eliminate the loss-metric inconsistency by introducing the differentiable rotational IoU algorithm. (2) In contrast to the parameter-based IoU losses, the TDIoU penalty term effectively reflects the overall difference between OBBs by measuring the distance between sampling points. In Table 4, to further investigate the effect of the angle parameter, we directly introduce it into the EIoU penalty term, which is named AIoU loss. However, AIoU loss is prone to non-convergence in the training phase, which is probably because the direct introduction of angle parameter will bring back the boundary discontinuity. On the contrary, the distance-based penalty term can reflect angle differences without directly employing the angle parameter. (3) Compared to the CDIoU loss, the introduction

of the centroid distance is able to speed up bounding box alignment. Figure 21 displays different regression loss curves in the training phase. The TDIoU loss directly minimizes the distance between corresponding centroids and vertices of two boxes and, thus, converges much faster than other losses. Moreover, since we use the triangle distance rather than the diagonal length of the smallest enclosing box to construct the denominator of penalty term, TDIoU loss reduces the training time by nearly half compared with other IoU-based losses, indicating that the computing complexity of our method is greatly reduced. All in all, the proposed TDIoU loss is more applicable to rotated bounding box regression.

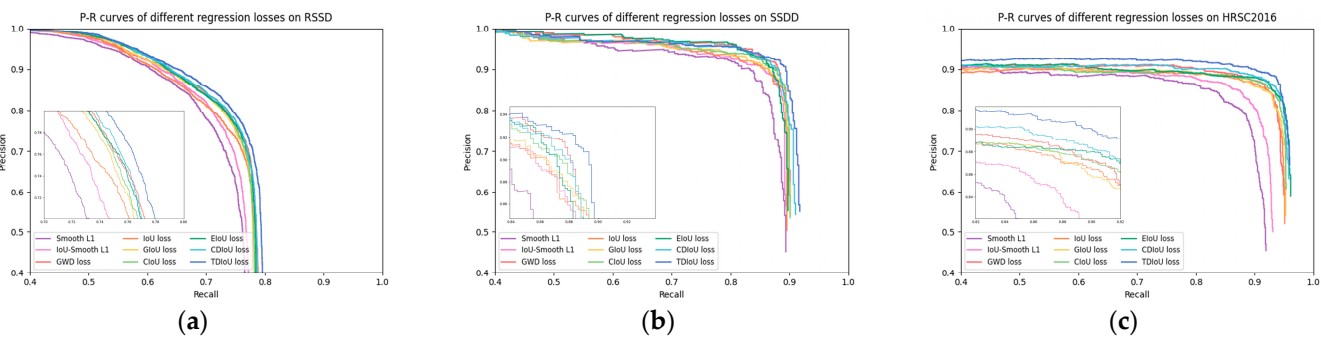

**Figure 20.** P–R curves of RetinaNet-R based on different regression losses on (**a**) RSSD, (**b**) SSDD, and (**c**) HRSC2016.

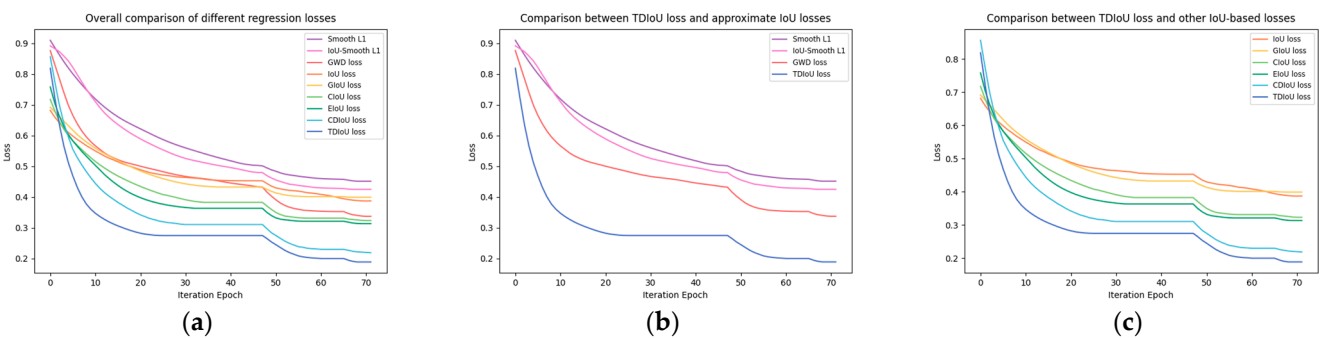

**Figure 21.** Regression loss curves of RetinaNet-R on RSSD. (**a**) Overall comparison of different regression losses; (**b**) comparison between TDIoU loss and approximate IoU losses; (**c**) comparison between TDIoU loss and other IoU-based losses.

6.3.3. Effectiveness of the AW-FPN

Since the proposed AW-FPN combines both multiple skip-scale connections and the attention-weighted feature fusion (AWF) strategy, we want to understand their respective contributions to accuracy improvement. Hence, we implement seven feature fusion networks with different connection pathways and fusion methods to verify the effectiveness of the AW-FPN, as shown in Tables 7–9. Notably, to eliminate the effect of irrelevant factors, the structure of all feature fusion networks is used only once.

Table 7 shows the results on our RSSD. The comparison between different connection pathways shows that the traditional FPN is inevitably limited by a single top-down information flow and achieves the lowest accuracy. The PANet with an extra bottom-up pathway improves by 0.68%, 0.59%, and 0.63% on inshore AP, offshore AP, and test AP, respectively. The BiFPN with single transverse skip-scale connections and the linear weighted fusion (LWF) strategy improves the AP by 1.87%, 1.13%, and 1.35%, respectively. For the AW-FPN with both transverse and longitudinal skip-scale connections, even the simplest additive fusion method can achieve performance similar to that of BiFPN. When using the same LWF method as BiFPN, the AW-FPN improves the AP by 2.38%, 1.34%, and 1.58%, indicating that longitudinal skip-scale connections are also crucial for feature fusion. For

comparisons between different fusion methods, AW-FPN improves by 2.91%, 1.63%, and 1.97% when using AFF (channel attention only) and by 4.09%, 2.06%, and 2.84% when using the proposed AWF (both channel and spatial attention), indicating that the attention-based fusion methods outperform the linear fusion methods and, that to generate non-linear fusion weights, it is better to use both channel and spatial attention rather than only using single channel attention. When CS$^2$A-Net is used as the base detector, our ultimate AW-FPN further improves the AP by 3.96%, 0.38%, and 1.50%. Similar experimental results are obtained on the other two datasets. From Tables 8 and 9, for the SSDD and HRSC2016, the proposed AW-FPN achieves the most outstanding performance on RetinaNet-R and a considerable improvement on the advanced CS$^2$A-Net, which proves the effectiveness of our approach.

**Table 7.** Comparison of different feature fusion networks on RSSD. The structure of all feature fusion networks is used only once in our experiment. Here, ADD represents the direct addition of feature maps, while LWF indicates the linear weighted fusion method in BiFPN [27].

| Detector | Fusion Network | Fusion Method | Fusion Type | Inshore AP | Offshore AP | Test AP |
|---|---|---|---|---|---|---|
| RetinaNet-R (R-50) | FPN (baseline) | ADD | Linear | 44.30 | 91.38 | 72.13 |
| | PANet [25] | ADD | Linear | 44.98 (+0.68) | 91.97 (+0.59) | 72.76 (+0.63) |
| | BiFPN [27] | LWF | Linear | 46.17 (+1.87) | 92.51 (+1.13) | 73.48 (+1.35) |
| | AW-FPN | ADD | Linear | 45.72 (+1.42) | 92.42 (+1.04) | 73.31 (+1.18) |
| | AW-FPN | LWF | Linear | 46.68 (+2.38) | 92.72 (+1.34) | 73.71 (+1.58) |
| | AW-FPN | AFF (channel) | Soft Selection | 47.21 (+2.91) | 93.01 (+1.63) | 74.10 (+1.97) |
| | AW-FPN | AWF (channel + spatial) | Soft Selection | **48.39 (+4.09)** | **93.44 (+2.06)** | **74.97 (+2.84)** |
| CS$^2$A-Net (R-50) | FPN (baseline) | ADD | Linear | 70.99 | 96.13 | 85.95 |
| | AW-FPN | AWF (channel + spatial) | Soft Selection | **74.95 (+3.96)** | **96.51 (+0.38)** | **87.45 (+1.50)** |

**Table 8.** Comparison of different feature fusion networks on SSDD.

| Detector | Fusion Network | Fusion Method | Fusion Type | Inshore AP | Offshore AP | Test AP |
|---|---|---|---|---|---|---|
| RetinaNet-R (R-50) | FPN (baseline) | ADD | Linear | 59.35 | 97.10 | 86.14 |
| | PANet [25] | ADD | Linear | 60.98 (+1.63) | 97.30 (+0.20) | 86.80 (+0.66) |
| | BiFPN [27] | LWF | Linear | 61.83 (+2.48) | 97.42 (+0.32) | 87.13 (+0.99) |
| | AW-FPN | ADD | Linear | 61.62 (+2.27) | 97.38 (+0.28) | 87.05 (+0.91) |
| | AW-FPN | LWF | Linear | 62.02 (+2.67) | 97.44 (+0.34) | 87.20 (+1.06) |
| | AW-FPN | AFF (channel) | Soft Selection | 63.11 (+3.76) | 97.54 (+0.44) | 87.58 (+1.44) |
| | AW-FPN | AWF (channel + spatial) | Soft Selection | **65.71 (+6.36)** | **97.91 (+0.81)** | **88.63 (+2.49)** |
| CS$^2$A-Net (R-50) | FPN (baseline) | ADD | Linear | 75.79 | 98.79 | 92.08 |
| | AW-FPN | AWF (channel + spatial) | Soft Selection | **78.83 (+3.04)** | **99.32 (+0.53)** | **93.38 (+1.30)** |

**Table 9.** Comparison of different feature fusion networks on HRSC2016.

| Detector | Fusion Network | Fusion Method | Fusion Type | Test AP$_{07}$ | Test AP$_{12}$ |
|---|---|---|---|---|---|
| RetinaNet-R (R-50) | FPN (baseline) | ADD | Linear | 81.63 | 84.82 |
| | PANet [25] | ADD | Linear | 82.44 (+0.81) | 85.62 (+0.80) |
| | BiFPN [27] | LWF | Linear | 82.74 (+1.11) | 85.92 (+1.10) |
| | AW-FPN | ADD | Linear | 82.50 (+0.87) | 85.68 (+0.86) |
| | AW-FPN | LWF | Linear | 82.86 (+1.23) | 86.30 (+1.48) |
| | AW-FPN | AFF (channel) | Soft Selection | 83.27 (+1.64) | 86.89 (+2.07) |
| | AW-FPN | AWF (channel + spatial) | Soft Selection | **84.10 (+2.47)** | **88.02 (+3.20)** |
| CS$^2$A-Net (R-50) | FPN (baseline) | ADD | Linear | 89.94 | 94.91 |
| | AW-FPN | AWF (channel + spatial) | Soft Selection | **90.21 (+0.27)** | **97.22 (+2.31)** |

Figure 22 shows the P–R curves of RetinaNet-R with different feature fusion networks. The P–R curve of AW-FPN is always higher than that of other methods. This may be because multiple skip-scale connections enhance semantic interactions between features of different resolutions and scales, which contributes to the complement of context information. In addition, in contrast to other linear fusion methods and the AFF using only channel attention, the proposed AWF aggregates global and local feature contexts in both the multi-scale channel attention module (MCAM) and the multi-scale spatial attention module (MSAM) to generate higher quality fusion weights. Figure 23 shows the feature visualization of different feature fusion networks. The region of interest (ROI) is highlighted in the feature heat map. The ROI in the feature maps generated by other methods is usually overlarge and contains considerable background clutter. In contrast, the contour and location of ships in the feature map generated by our AW-FPN is more distinct and accurate, which helps the detectors to focus more on the real ship targets rather than background clutter, and to learn more useful context information.

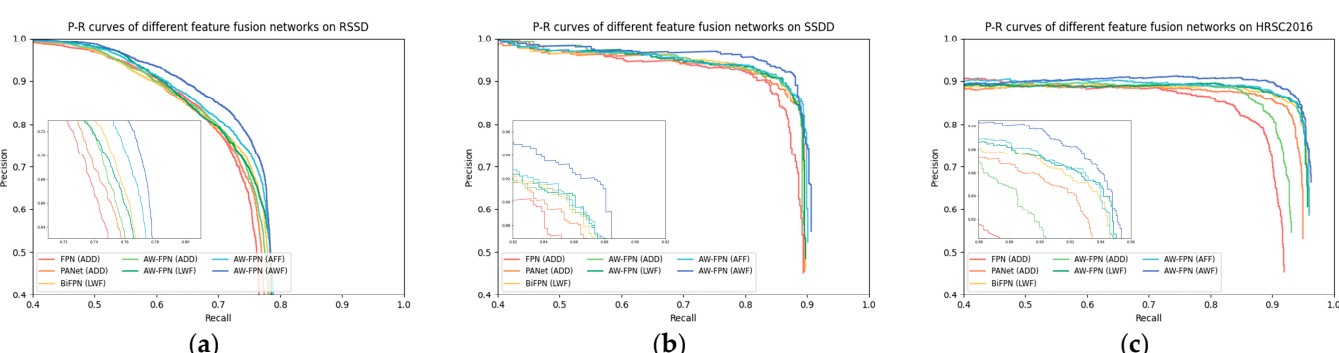

**Figure 22.** P–R curves of RetinaNet-R with different feature fusion networks on (**a**) RSSD, (**b**) SSDD, and (**c**) HRSC2016.

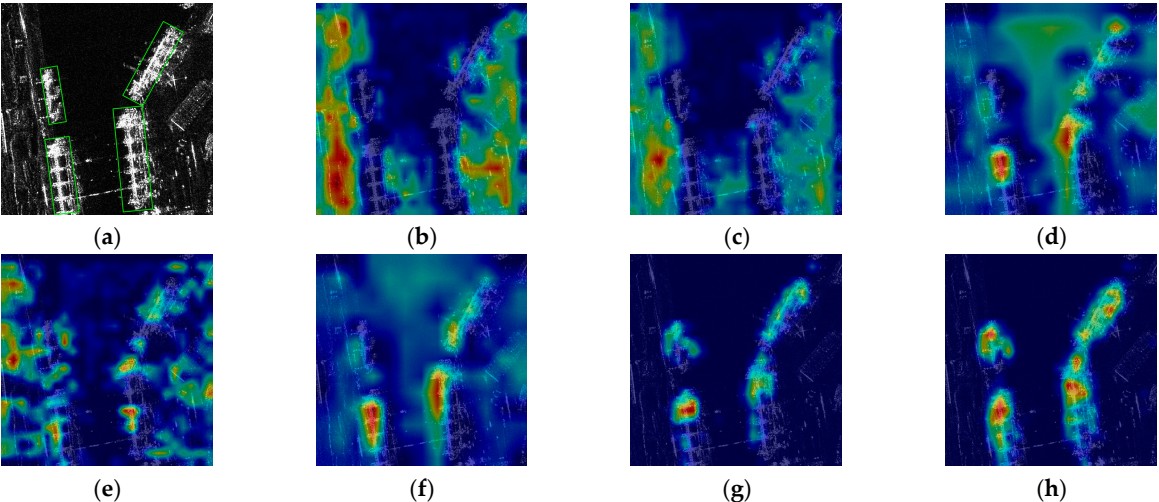

**Figure 23.** Feature visualization of different feature fusion networks (take $P_5$ as an example). (**a**) Input; (**b**) FPN (ADD); (**c**) PANet (ADD); (**d**) BiFPN (LWF); (**e**) AW-FPN (ADD); (**f**) AW-FPN (LWF); (**g**) AW-FPN (AFF); (**h**) AW-FPN (AWF).

### 6.4. Comparison with the State-of-the-Art

We embed the proposed AW-FPN into CS$^2$A-Net and train it with our TDIoU loss. and then compare our approach with the state-of-the-art methods on three datasets.

#### 6.4.1. Results on the RSSD

Table 10 provides a quantitative comparison of different methods on RSSD. As can be seen, the latest two-stage detection methods, such as CSL, SCRDet++, and ReDet, generally

achieve outstanding performance. However, they always adopt complex structures in exchange for improved accuracy at the expense of detection efficiency. Lately, some single-stage detection methods, such as R$^3$Det, GWD, and CS$^2$A-Net, have been presented, which show competitive performance and efficiency on RSSD. Our method can further improve the accuracy of these rotation detectors and has a minimal impact on detection efficiency. As per Table 10, the proposed approach achieves 75.41%, 96.62%, and 87.87% accuracy in terms of inshore AP, offshore AP, and test AP on CS$^2$A-Net, respectively, without using multi-scale training and testing, which is already extremely close to the performance of the advanced ReDet and GWD. When employing a stronger backbone (i.e., ResNet101) and multi-scale training and testing, our approach achieves state-of-the-art performance, with the AP of 77.65%, 97.35%, and 89.18%, respectively, which is 1.98%, 0.65%, and 1.11% higher than that of the suboptimal method (i.e., GWD). Furthermore, the inference speed of our method reaches 12.1 fps, which is 11.1 fps and 2.5 fps faster than that of ReDet and GWD, respectively. Compared to the original CS$^2$A-Net, our approach trades off a speed loss of only 1.1 fps for significant gains, of 3.48%, 0.87%, and 1.85%, in accuracy.

**Table 10.** Comparison with state-of-the-art methods on RSSD. Here, **MS** indicates the multi-scale training and testing, **FPS** is obtained by calculating the overall inference time and the number of images, TDIoU + AW-FPN represents the CS$^2$A-Net detector based on TDIoU loss and AW-FPN, R-50 refers to ResNet50 (likewise R-101, R-152), and ReR-50 and H-104 denote ReResNet50 [68] and Hourglass104, respectively [69].

| Method | Backbone | Stage | MS | Inshore AP | Offshore AP | Test AP | FPS |
|---|---|---|---|---|---|---|---|
| SCRDet [41] | R-101 | Two | ✓ | 65.47 | 95.53 | 83.65 | 5.0 |
| RSDet [70] | R-152 | Two | | 68.48 | 95.88 | 84.85 | – |
| Gliding Vertex [71] | R-101 | Two | | 70.83 | 96.11 | 85.80 | – |
| CSL [44] | R-152 | Two | ✓ | 71.45 | 96.18 | 86.15 | 4.0 |
| SCRDet++ [72] | R-101 | Two | ✓ | 71.66 | 96.21 | 86.24 | 5.0 |
| ReDet [68] | ReR-50 | Two | ✓ | 75.57 | 96.65 | 88.03 | <1.0 |
| RetinaNet-R [17] | R-50 | Single | | 44.30 | 91.38 | 72.13 | 17.5 |
| DRN [73] | H-104 | Single | ✓ | 67.95 | 95.68 | 84.40 | – |
| R$^3$Det [74] | R-152 | Single | ✓ | 71.47 | 96.25 | 86.21 | 9.6 |
| DCL [45] | R-101 | Single | ✓ | 71.92 | 96.23 | 86.36 | 12.0 |
| GWD [23] | R-152 | Single | ✓ | 75.67 | 96.70 | 88.07 | 9.6 |
| CS$^2$A-Net [67] | R-50 | Single | | 70.99 | 96.13 | 85.95 | 16.5 |
| CS$^2$A-Net [67] | R-101 | Single | ✓ | 74.17 | 96.48 | 87.33 | 13.2 |
| TDIoU+AW-FPN (ours) | R-50 | Single | | 75.41 | 96.62 | 87.87 | 15.1 |
| TDIoU + AW-FPN (ours) | R-101 | Single | ✓ | 77.65 | 97.35 | 89.18 | 12.1 |

Figure 24 shows qualitative results of different methods on RSSD. As per the results of the offshore scene containing multiple ships (the first row), the other four methods detect islands and reefs incorrectly as ships, while our method is more robust in distinguishing small ships from background components. For the complex inshore scenes (the second row to the fourth row), the results of other methods include false alarms and leave some vessels undetected. In contrast, our method succeeds in detecting all ships and locating them more precisely, especially for densely arranged ships close to man-made facilities.

### 6.4.2. Results on the SSDD

Table 11 shows experimental results of different methods on the SSDD. Since SSDD contains few SAR images and the scenes are relatively simple, the ship detection accuracy is generally high. As shown in Table 11, based on CS2A-Net (R-50), our approach achieves 80.75%, 99.64%, and 94.05% of inshore AP, offshore AP, and test AP, respectively. When using ResNet101 as the backbone network, the AP of our approach reaches 84.34%, 99.71%, and 95.16%, compared to the state-of-the-art detectors ReDet (82.80%, 99.18%, and 94.27%) and GWD (81.99%, 99.66%, and 94.35%). Moreover, our approach improves the overall accuracy by 1.26% and 0.44% compared to BiFA-YOLO and R2FA-Det, respectively, and the inference speed by 5.5 fps compared to the suboptimal R2FA-Det, indicating that the proposed method achieves the best performance and satisfies high detection efficiency.

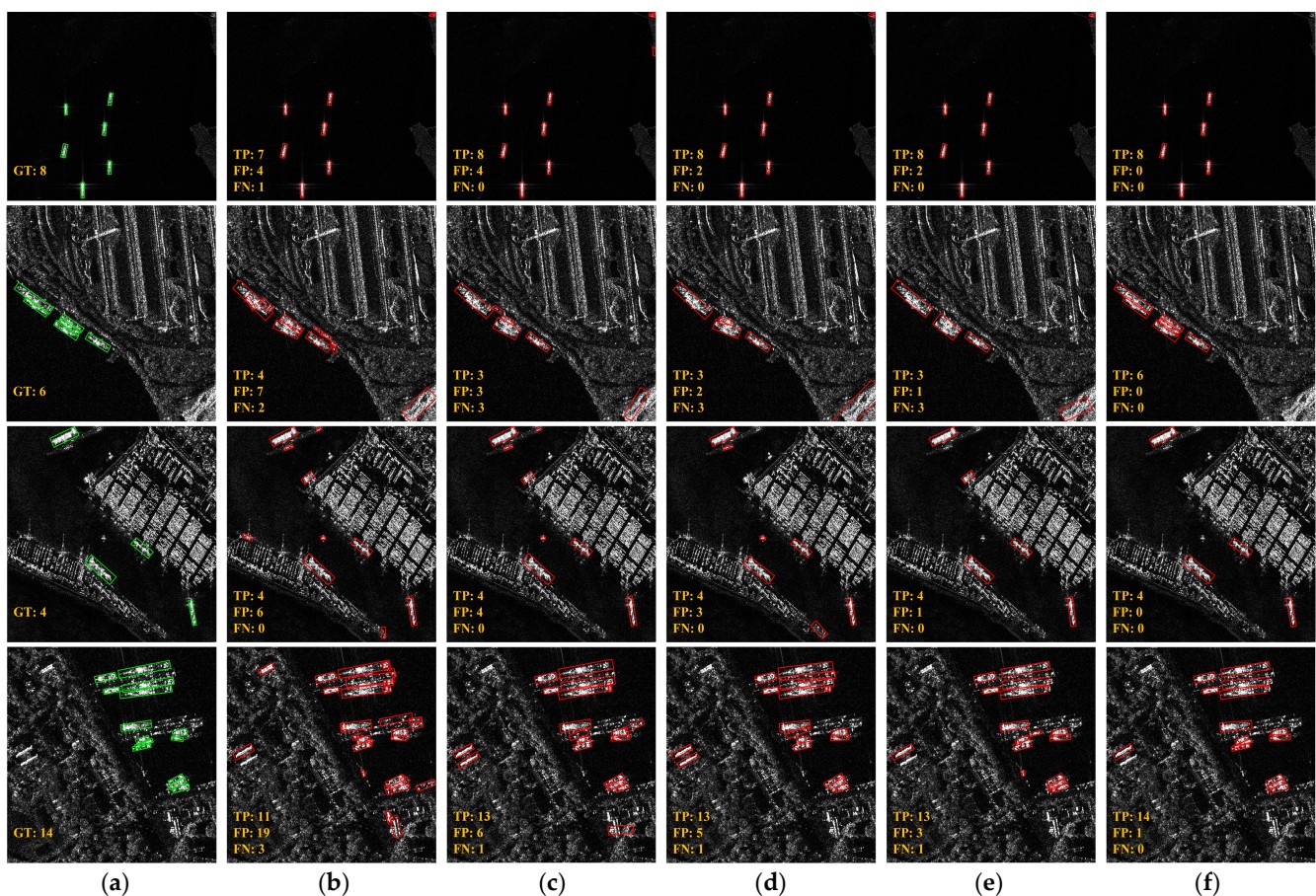

(a)       (b)       (c)       (d)       (e)       (f)

**Figure 24.** Detection results of different methods on RSSD. (**a**) Ground truth (GT); (**b**) RetinaNet-R; (**c**) CS$^2$A-Net; (**d**) ReDet; (**e**) GWD; (**f**) TDIoU + AW-FPN (ours). Green and red boxes represent real ship targets and detection results, respectively.

**Table 11.** Comparison with state-of-the-art methods on SSDD. Here, V-16 and C-53 denote VGG16 [75] and CSPDarknet53 [76]. The method with * indicates that its results are from the corresponding paper. Here, (<800) indicates that the long side of images is less than 800 pixels.

| Method | Backbone | Stage | Image Size | Inshore AP | Offshore AP | Test AP | FPS |
|---|---|---|---|---|---|---|---|
| Cascade RCNN * [19] | R-50 | Multiple | 300 × 300 | – | – | 88.45 | 2.8 |
| MSR2N * [16] | R-50 | Two | (<800) × 350 | – | – | 90.11 | 9.7 |
| Gliding Vertex [71] | R-101 | Two | 512 × 512 | 75.23 | 98.35 | 91.88 | – |
| CSL [44] | R-152 | Two | 512 × 512 | 76.15 | 98.87 | 92.16 | 7.0 |
| SCRDet + + [72] | R-101 | Two | 512 × 512 | 77.17 | 99.16 | 92.56 | 8.8 |
| ReDet [68] | ReR-50 | Two | 512 × 512 | 82.80 | 99.18 | 94.27 | <1.0 |
| RetinaNet-R [17] | R-50 | Single | 512 × 512 | 59.35 | 97.10 | 86.14 | **30.6** |
| R$^3$Det [74] | R-152 | Single | 512 × 512 | 76.92 | 99.09 | 92.29 | 16.9 |
| DRBox-v2 * [77] | V-16 | Single | 300 × 300 | – | – | 92.81 | 18.1 |
| BiFA-YOLO * [22] | C-53 | Single | 512 × 512 | – | – | 93.90 | – |
| GWD [23] | R-152 | Single | 512 × 512 | 81.99 | 99.66 | 94.35 | 16.9 |
| R$^2$FA-Det * [19] | R-101 | Single | 300 × 300 | – | – | 94.72 | 15.8 |
| CS$^2$A-Net [67] | R-50 | Single | 512 × 512 | 75.79 | 98.79 | 92.08 | 29.0 |
| CS$^2$A-Net [67] | R-101 | Single | 512 × 512 | 79.01 | 99.41 | 93.47 | 23.2 |
| TDIoU + AW-FPN (ours) | R-50 | Single | 512 × 512 | 80.75 | 99.64 | 94.05 | 26.6 |
| TDIoU + AW-FPN (ours) | R-101 | Single | 512 × 512 | 84.34 | 99.71 | 95.16 | 21.3 |

Figure 25 visualizes some detection results of different methods on the SSSD. In the complex inshore scenes, the other three methods suffer from missed and false detection under background clutter interference. In contrast, our approach is highly robust and displays superiority in detecting densely distributed small ships.

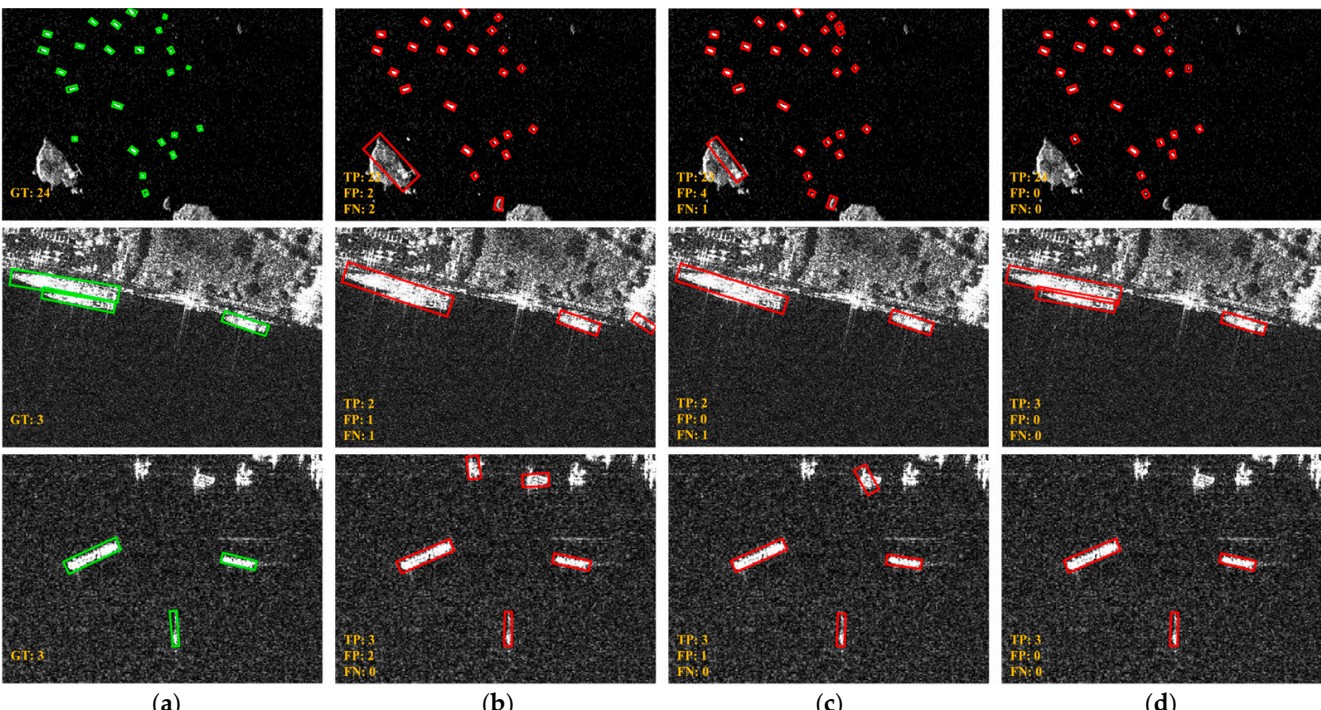

**Figure 25.** Detection results of different methods on SSDD. (**a**) GT; (**b**) CS$^2$A-Net; (**c**) GWD; (**d**) TDIoU + AW-FPN (ours).

### 6.4.3. Results on the HRSC2016

To verify the effectiveness and robustness of our approach in optical remote sensing scenarios, we conduct experiments with state-of-the-art methods on the HRSC2016, which contains a great number of ships with large aspect ratios and arbitrary orientations. As shown in Table 12, our approach achieves 90.71% and 98.65% accuracy on the metrics AP$_{07}$ and AP$_{12}$, respectively, outperforming other comparison methods. Compared with the suboptimal approach (i.e., ReDet), the proposed method improves the accuracy by 0.25% and 1.02%. In addition, the inference speed of our method is 16.9 fps, which is much faster than that of the two-stage method ReDet (<1.0 fps). As per the above results, our method shows excellent generalization ability in other rotation detection scenarios.

**Table 12.** Comparison with state-of-the-art methods on HRSC2016. The method with * indicates that its results are from the corresponding paper.

| Method | Backbone | Stage | Image Size | Test AP$_{07}$ | Test AP$_{12}$ | FPS |
|---|---|---|---|---|---|---|
| RoI-Transformer * [78] | R-101 | Two | 800 × 512 | 86.20 | – | 6.0 |
| RSDet * [70] | R-50 | Two | 800 × 800 | 86.50 | – | – |
| Gliding Vertex * [71] | R-101 | Two | – | 88.20 | – | – |
| CenterMap-Net * [79] | R-50 | Two | – | – | 92.80 | – |
| CSL [44] | R-101 | Two | 800 × 800 | 89.62 | 96.10 | 5.0 |
| ReDet [68] | ReR-50 | Two | 800 × 512 | 90.46 | 97.63 | <1.0 |
| RetinaNet-R [17] | R-50 | Single | 800 × 512 | 81.63 | 84.82 | **24.4** |
| DRN * [73] | H-104 | Single | – | – | 92.70 | – |
| R$^3$Det [74] | R-101 | Single | 800 × 800 | 89.26 | 96.01 | 12.0 |
| DCL [45] | R-101 | Single | 800 × 800 | 89.46 | 96.41 | 12.0 |
| GWD [23] | R-101 | Single | 800 × 800 | 89.85 | 97.37 | 12.0 |
| CS$^2$A-Net [67] | R-50 | Single | 800 × 512 | 89.94 | 94.91 | 23.0 |
| CS$^2$A-Net [67] | R-101 | Single | 800 × 512 | 90.17 | 95.01 | 18.4 |
| TDIoU + AW-FPN (ours) | R-50 | Single | 800 × 512 | 90.35 | 97.54 | 21.1 |
| TDIoU + AW-FPN (ours) | R-101 | Single | 800 × 512 | 90.71 | 98.65 | 16.9 |

To evaluate the capability of our method to detect ships with extreme aspect ratios, we choose three images containing ships with large aspect ratios. As shown in Figure 26, our approach has fewer false alarms than any other methods. In addition, the position and

orientation of the predicted box generated by our method are much closer to those of the ground truth.

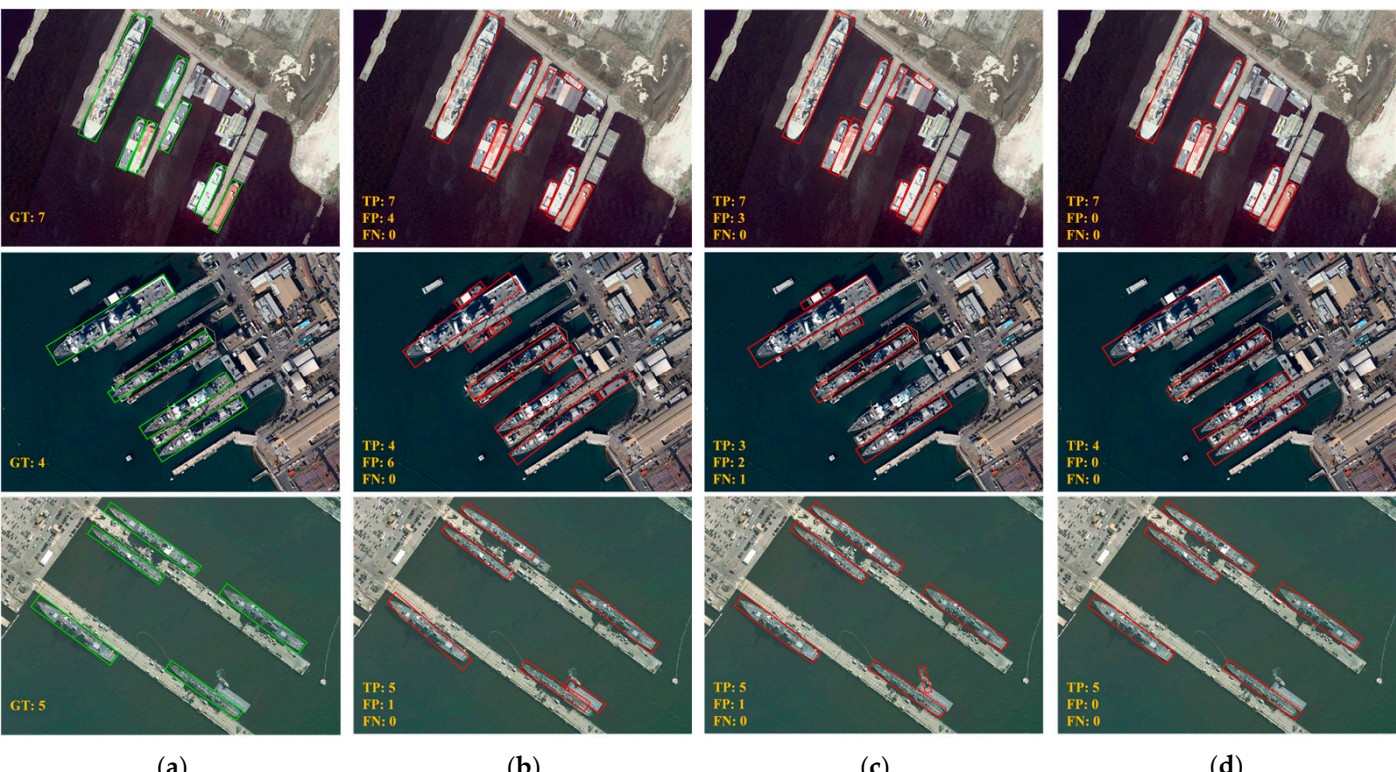

**Figure 26.** Detection results of different methods on HRSC2016. (**a**) GT; (**b**) CS$^2$A-Net; (**c**) GWD; (**d**) TDIoU + AW-FPN (ours).

Figure 27 displays P–R curves of different methods on RSSD, SSDD, and HRSC2016. It can be found that the P–R curve of our method is almost always higher than those of the other methods. Through all the above experiments and discussions, we can draw the conclusion that the proposed TDIoU loss and AW-FPN can improve the detection accuracy of arbitrary-oriented ships in both SAR scenes and optical remote sensing scenes, especially in the case of extreme scale and aspect ratio variations. This may be attributed to the fact that TDIoU loss fundamentally eliminates the loss-metric inconsistency and angular boundary discontinuity, so as to guide the rotation detector to achieve more accurate boundary box regression. Furthermore, the proposed AW-FPN is improved in terms of both the connection pathway and the fusion method, enabling high-quality semantic interactions and soft feature selections between features of inconsistent resolutions and scales.

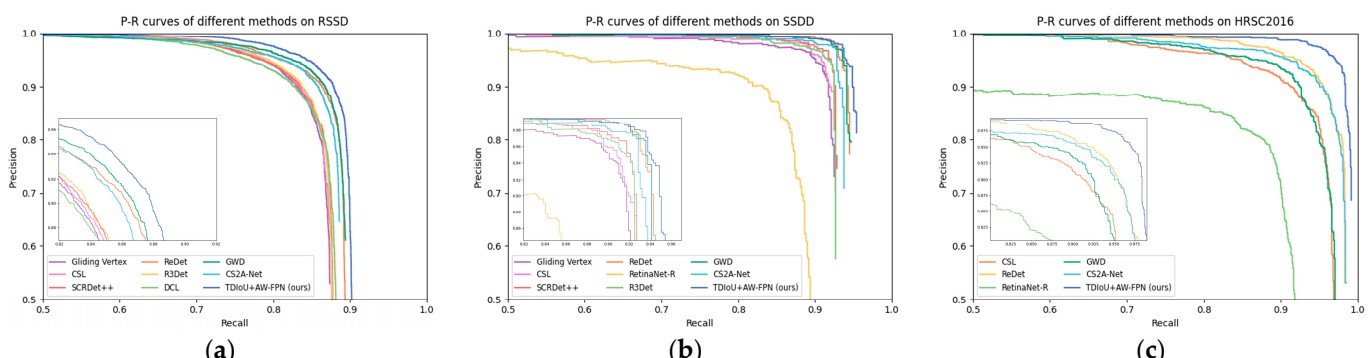

**Figure 27.** P–R curves of different methods on (**a**) RSSD, (**b**) SSDD and (**c**) HRSC2016.

## 7. Conclusions

In this paper, a unified framework combining TDIoU loss, AW-FPN, and RSSD is proposed to improve the capability of rotation detectors in recognizing and locating ships in SAR images. (1) The rotational IoU algorithm based on the Shoelace formula opens up the possibility of using IoU-based loss for rotated bounding box regression. On this basis, an effective TDIoU penalty term is designed to overcome the defects of existing IoU-based losses and solve the problems caused by angle regression. (2) Here, AW-FPN improves previous methods from connection pathways and fusion methods. Skip-scale connections enhance semantic interactions between multi-scale features. The AWF generates attention fusion weights via MCAM and MSAM to encode emphasized and suppressed positions in feature maps, making detectors focus more on real ship targets. (3) We construct a challenging benchmark, namely RSSD, for arbitrary-oriented SAR ship detection. Ships in RSSD not only differ significantly in orientations but also features multi-scale characteristics. In addition, 15 baseline results are provided for research. (4) Extensive experiments are conducted on three datasets. When using TDIoU loss and AW-FPN, even the advanced $CS^2A$-Net is able to improve upon the AP by 1.85%, 1.69%, and 0.54% on RSSD, SSDD, and HRSC2016, respectively, fully demonstrating the effectiveness and robustness of our approach.

Our future work is summarized as follows:

1. Though numerous innovative methods have emerged in SAR ship detection, due to the limitation of datasets, most of them are still based on HBBs. Therefore, we will further improve our TDIoU loss and AW-FPN, and try to combine them with more advanced rotation detection methods to improve the detection accuracy of arbitrary-oriented ships, especially in complex inshore scenes;

2. We will keep maintaining and updating RSSD to v2.0 or higher. Specifically, this will involve increasing the number of ship slices, incorporating more diverse SAR scenarios, building more standardized baselines, providing more accurate polygon annotations, etc. In the near future, it will be publicly available to facilitate further research in this field.

3. We will explore the possibility of multi-classification of ship targets in SAR images, which is an emerging research topic. With the development of high-resolution SAR image generation technology, the category information will be integrated into ship detection, which is beneficial for the progress of SAR intelligent interpretation technology.

**Author Contributions:** Conceptualization, R.G.; methodology, R.G.; software, R.G.; validation, R.G.; formal analysis, R.G.; investigation, R.G. and Z.X.; resources, R.G. and Z.X.; data curation, R.G., Z.X. and Q.X.; writing—original draft preparation, R.G.; writing—review and editing, R.G., Z.X., K.H. and Q.X.; visualization, R.G.; supervision, Z.X. and K.H.; project administration, R.G.; funding acquisition, Z.X. and K.H. All authors have read and agreed to the published version of the manuscript.

**Funding:** This research was funded by National Key Research and Development Program, grant number 2019YFB1600605; The Youth Fund from National Natural Science Foundation of China, grant number 62101316; Shanghai Sailing Program, grant number 20YF1416700.

**Data Availability Statement:** Not applicable.

**Conflicts of Interest:** The authors declare no conflict of interest.

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
