# Peer review of "Triangle Distance IoU Loss, Attention-Weighted Feature Pyramid Network, and Rotated-SARShip Dataset for Arbitrary-Oriented SAR Ship Detection"

_remotesensing, doi:10.3390/rs14184676_

Round 1

Reviewer 1 Report

Summary:

This is an interesting work.

The recognition of objects in SAR images is a hot topic in this direction in recent years. However, there is still a lot of research space and practical significance on how to accurately recognize small targets similar to ships in SAR images.

In order to accurately identify and locate ships with different dimensions, large aspect ratio, dense arrangement and arbitrary direction,In this manuscript, a unified framework is proposed, which combines triangle range IOU loss , attention weighted feature pyramid network, and rotating SAR ship data set for SAR ship detection in any direction. The recognition accuracy of ships in any direction is effectively improved.

The present form of the manuscript is well written, although some minor problems need to be addressed.

The paper can be improved in the following aspects:

Detailed comments:

1. In section 4,you said“How to calculate the area of the intersection area is the core issue.” Can you explain which reason leads this core issue?

2. In section 4, I think it is unnecessary to write out the final calculation results for equations (10) and (11).

3. I think you can use more specific diagrams or tables to describe the network structure used.

4. I suggest emphasizing the importance of MCAM and MSAM before introducing them.

5. In section 6, the HRSC2016 dataset has been improved with the help of TDIOU loss in terms of the 2007 and 2012 evaluation metrics.Are there any new evaluation metrics of this dataset in recent years?

6. I suggest rearranging the pictures to make the details in the pictures more clear.

Reviewer 2 Report

The paper proposed a unified framework based on deep learning for Ship detection in SAR scenes.  Several innovations and modifications to the structure of deep learning-based object detectors have been developed that make the designed framework capable to outperform the state-of-the-art ship detectors. The manuscript first discusses the deficiencies of algorithms previously implemented in the literature for ship detection and then proposes some solutions for accurately recognizing and locating ships of varying scales, large aspect ratios, dense arrangements, and arbitrary orientations in SAR images.  In more detail, the proposed framework integrates triangle distance IoU loss (TDIoU loss), an attention-weighted feature pyramid network (AW-FPN), and a Rotated-SARShip dataset (RSSD) for ship detection in SAR images.

The paper has been organized well. Deficiencies in previous work have been discussed comprehensively and innovative ideas have been implemented for improving the capability of arbitrary-oriented ship detection.  The reviewer thinks that the paper has sufficient novelty and quality for publication. However, some minor comments are listed below:

Line 44-45: “most of them simply migrate ...”, The sentence is not clear to understand. Maye “migrate” is not a suitable verb for your description. 

Figure 1: It would be better to bring the names of algorithms for ship detection in images (a-d).

Line 62: Some references are required after the sentence “oriented bounding box (OBB)-based methods have emerged”.

Figure 2: What Pi denote? It should be clarified in the caption of the figure or the context of the paper.

Table 1: Are there any advantages of Polygon-based detection vs OBB? It is recommended to explain more about polygon-based object detection. In the context of the paper, you have nicely explained the deficiencies of HBB-based ship detection but no explanation has been provided for polygon-based annotation and detection.

Figure 4: As you described, the TDIoU loss removes the issues of normal IoU. It would be very nice to visualize the relationship between TDIoU loss and angle differences and aspect ratios (similar to figure 4).

Figure 16: Bring the ground truth annotations to make the comparison possible.

Reviewer 3 Report

The authors present a unified framework for arbitrary-oriented SAR ship detection from three aspects: loss function, network structure, and training data. To address the loss-metric inconsistency and boundary discontinuity, a triangle distance IoU loss (TDIoU loss) combining a differentiable rotational IoU algorithm is proposed. Then an attention-weighted feature pyramid network (AW-FPN) is proposed to improve conventional FPNs from the connection pathway and the fusion method. Finally, a challenging Rotated-SARShip dataset (RSSD) is constructed.

The content of this paper is sufficient and the idea is interesting. Comprehensive data analysis and extensive experimental results have been provided. The manuscript could be published with some modifications. Here are some points that could be addressed by the authors to improve their work.

1. There are some typos in the manuscript that should be modified. I suggest using a spell checker.

2. There are a few of acronyms are not defined in the text, please double check and make corrections.

3. The conclusion need to be improved. I suggest extending the conclusions section to focus on the experimental results and future work on the proposed method and dataset.

Round 2

Reviewer 1 Report

Summary:

This is an interesting work. 

The recognition of objects in SAR images is a hot topic in this direction in recent years. However, there is still a lot of research space and practical significance on how to accurately recognize small targets similar to ships in SAR images.

In order to accurately identify and locate ships with different dimensions, large aspect ratio, dense arrangement and arbitrary direction, in this manuscript, a unified framework is proposed, which combines triangle range IOU loss, attention weighted feature pyramid network, and rotating SAR ship data set for SAR ship detection in any direction. The recognition accuracy of ships in any direction is effectively improved.

For the revised manuscript, the following comments are made:

Detailed comments:

1.The section 4, has reorganized the sentences and explained“ How to calculate the area of the intersection area is the core issue.” This makes the focus of this section easier to be understanded.

2.The formulations of equations (10) and (11) in the first manuscript was modified to make the reasoning process more concise.

3.In section 6, Added a detailed structure diagram to describe and explain the two networks used, and replied to the comment 3.

4.In response to the last comment 4, after modification, MCAM and MSAM are briefly explained in the section 2, so that readers can understand the importance of them.

5.In section 6, the conclusion part is the same as the previous edition, but it is more rigorous in language. Relevant background is added before the conclusion so that readers can perceive the effectiveness of the proposed method when reading.

6.In section 7, future work is added, and the summary language is more concise. It is convenient for readers to further study and research.

Reviewer 2 Report

It is OK for publication.